# On the formation of highly oxidized pollutants by autoxidation of terpenes under low temperature combustion conditions: the case of limonene and α-pinene.

Roland Benoit[1], Nesrine Belhadj[1,2], Zahraa Dbouk[1,2], Maxence Lailliau[1,2], and Philippe Dagaut[1]

[1]CNRS-INSIS, ICARE, Orléans, France, roland.benoit@cnrs-orleans.fr, nesrine.belhadj@cnrs-orleans.fr, zahraa.dbouk@cnrs-orleans.fr, maxence.lailliau@cnrs-orleans.fr, dagaut@cnrs-orleans.fr

[2]Université d'Orléans, Orléans, France

**Correspondance**: Roland Benoit (roland.benoit@cnrs-orleans.fr)

**Abstract.**

The oxidation of monoterpenes under atmospheric conditions has been the subject of numerous studies. They were motivated by the formation of oxidized organic molecules (OOM) which, due to their low vapor pressure, contribute to the formation of secondary organic aerosols (SOA). Among the different reaction mechanisms proposed for the formation of these oxidized chemical compounds, it appears that the autoxidation mechanism, involving successive events of $O_2$ addition and H-migration, common to both low-temperature combustion and atmospheric conditions, is leading to the formation of highly oxidized products (HOPs). However, cool flame oxidation (~500-800 K) of terpenes has not received much attention even if it can contribute to atmospheric pollution through biomass burning and wildfires. Under such conditions, terpenes can be oxidized via autoxidation. In the present work, we performed oxidation experiments with limonene-oxygen-nitrogen and α-pinene-oxygen-nitrogen mixtures in a jet-stirred reactor (JSR) at 590 K, a residence time of 2 s, and atmospheric pressure. Oxidation products were analyzed by liquid chromatography, flow injection, and soft ionization-high resolution mass spectrometry. H/D exchange and 2,4-dinitrophenyl hydrazine derivatization were used to assess the presence of OOH and C=O groups in oxidation products, respectively. We probed the effects of the type of ionization used in mass spectrometry analyses on the detection of oxidation products. Heated electrospray ionization (HESI) and atmospheric pressure chemical ionization (APCI), in positive and negative modes were used. We built an experimental database consisting of literature data for atmospheric oxidation and presently obtained combustion data for the oxidation of the two selected terpenes. This work showed a surprisingly similar set of oxidation products chemical formulas, including oligomers, formed under the two rather different conditions, i.e., cool flame and simulated atmospheric oxidation. Data analysis (in HESI mode) indicated that a subset of chemical formulas is common to all experiments, independently of experimental conditions. Finally, this study indicates that more than 45% of the detected chemical formulas in this full dataset can be ascribed to an autoxidation reaction.

## 1 Introduction

Terpenes are emitted into the troposphere by vegetation (Seinfeld and Pandis, 2006). They can be used as drop in fuels (Harvey et al., 2010;Mewalal et al., 2017;Harvey et al., 2015) which could increase emissions via fuel evaporation and unburnt fuel release. Biomass burning and wildfires can also release terpenes and their products of oxidation into the troposphere (Gilman et al., 2015;Hatch et al., 2019;Schneider et al., 2022). Wildfires temperature ranges from 573 to 1373 K (Wotton et al., 2012), which covers both the cool flame (~500-800 K) and intermediate to high temperature combustion regimes. Products of biomass burning have been characterized earlier (Smith et al., 2009). Using van Krevelen diagrams, the authors reported H/C versus O/C in the ranges 0.5 to 3 and 0 to 1, respectively. Whereas a large fraction of these products can derive from cellulose, hemicellulose, and lignin oxidation, their formation via terpenes oxidation cannot be ruled out. In a more recent study (Gilman et al., 2015), it was reported that biomass burning emissions were dominated by oxidized organic compounds (57 to 68% of total mass emissions). Wildfires are getting more and more frequent and their intensity increases(Burke et al., 2021). In large wildfires, there are many updrafts which can transport a variety of materials ranging from gases to particulates, and even bacteria (Kobziar et al., 2018). Furthermore, it was recently demonstrated that recent wildfires in Australia produced smoke which could reach an altitude of 35 km (Khaykin et al., 2020). Such events could contribute to ozone destruction (Bernath et al., 2022) but also to tropospheric pollution. But, field measurements are not appropriate for comparison with the present data because a strict distinction on the origins of the chemical compounds observed cannot be assessed. For example, literature works and reviews (Hu et al., 2018;Popovicheva et al., 2019;Prichard et al., 2020) present field measurements from smoldering fires which were not detailed enough to be used here.

Cool flame oxidation is dominated by autoxidation (Bailey and Norrish, 1952;Benson, 1981;Cox and Cole, 1985;Korcek et al., 1972) which involves peroxy radicals (ROO$^{\bullet}$). Autoxidation is based on an H-shift and oxygen addition which starts with the initial production of ROO$^{\bullet}$ radicals. This mechanism can repeat itself several times and lead to recurrent oxygen additions to form highly oxidized products (Wang et al., 2017;Wang et al., 2018;Belhadj et al., 2020;Belhadj et al., 2021a;Belhadj et al., 2021b):

R$\bullet$+ O$_2$ $\rightleftarrows$ ROO$\bullet$ (first O$_2$-addition)

ROO$\bullet$ $\rightleftarrows$$\bullet$QOOH (H-shift)

$\bullet$QOOH +O$_2$ $\rightleftarrows$ $\bullet$OOQOOH (second O$_2$-addition)

$\bullet$OOQOOH $\rightleftarrows$ HOOQ$\bullet$'OOH (H-shift)

HOOQ$\bullet$'OOH +O$_2$ $\rightleftarrows$ (HOO)$_2$Q'OO$\bullet$ (third O$_2$-addition)

(HOO)$_2$Q'OO$\bullet$ $\rightleftarrows$ (HOO)$_2$Q$\bullet$''OOH (H-shift)

(HOO)$_2$Q$\bullet$''OOH +O$_2$ $\rightleftarrows$ (HOO)$_3$Q''OO$\bullet$ (fourth O-addition), etc.

There, the formation of highly oxidized products (HOPs) was mainly attributed to autoxidation reactions (Belhadj et al., 2021c;Benoit et al., 2021).

In atmospheric chemistry, it is only relatively recently that this pathway has been considered (Vereecken et al., 2007;Crounse et al., 2013;Jokinen et al., 2014a;Ehn et al., 2014;Berndt et al., 2015;Jokinen et al., 2015;Berndt et al., 2016;Iyer et al., 2021). Also, it has been identified that highly oxygenated organic molecules (HOMs), a source of secondary organic aerosols (SOA), can result from autoxidation processes (Ehn et al., 2014;Wang et al., 2021;Tomaz et al., 2021;Bianchi et al., 2019). Modeling studies complemented by laboratory experiments showed that autoxidation

mechanisms proceed simultaneously on different ROO˙ radicals leading to the production of a wide range of oxidized
compounds in a few hundredths of a second (Jokinen et al., 2014a;Berndt et al., 2016;Bianchi et al., 2019;Iyer et al.,
2021). Recent works have shown that, under certain atmospheric conditions, this autoxidation mechanism could be
competitive with other reaction pathways involving ROO˙ radicals (Bianchi et al., 2019), e.g., the carbonyl channel
(ROO˙ → $R_H$O + OH), the hydroperoxide channel (ROO˙ + HOO˙ → ROOH + $O_2$ and RO˙ + ˙OH + $O_2$),
disproportionation reactions (ROO˙ + R'OO˙ → RO˙ + R'O˙ + $O_2$ and $R_H$O + R'OH + $O_2$), accretion reactions (ROO˙
+ R'OO˙ → ROOR' + $O_2$). Similarity, in terms of observed chemical formulas of products from cool flame oxidation
of limonene and atmospheric oxidation of limonene, has been reported recently (Benoit et al., 2021). The same year,
Wang et al. showed that the oxidation of alkanes follows this autoxidation mechanism under both atmospheric and
combustion conditions (Wang et al., 2021). Also, that work confirmed that internal H-shifts in autoxidation can be
promoted by the presence of functional groups, as predicted earlier (Otkjær et al., 2018) for ROO˙ radicals containing
OOH, OH, $OCH_3$, $CH_3$, C=O, or C=C groups. Autoxidation will preferentially form chemical functions such as
carbonyls, hydroperoxyl, or peroxyl. This large diversity of chemical functions will promote the formation of isomers.
Nevertheless, the common point to these chemical compounds is the sequential addition of $O_2$. Therefore, in a database,
potential candidate products of autoxidation are easily identified by this sequential addition.
To better understand the importance of these reaction pathways, the experimental conditions unique to these two
chemistries must be considered. In laboratory studies conducted under simulated atmospheric conditions, oxidation
occurs at near-ambient temperatures (250-300 K), at atmospheric pressure, in the presence of ozone and/or ˙OH
radicals (Table S1), used to initiate oxidation with low initial terpene concentrations. In combustion, the ˙OH radical,
temperature, and pressure are driving autoxidation. In addition to the increase in temperature, the initial concentrations
of the reagents are generally higher compared to the atmospheric conditions, in order to initiate the oxidation with $O_2$,
which is much slower than that involving ozone or ˙OH. Rising temperature increases isomerization rates and favors
autoxidation, at the expense of other possible reactions of ROO˙ radicals. Indeed, it has been reported earlier that a
temperature rise from 250 to 273K does not affect the distribution of HOMs (Quéléver et al., 2019) whereas Tröstl et
al. suggested that the distribution of HOMs is affected by temperature, α-pinene or particle concentration (Tröstl et al.,
2016). Similarly, the experiments of Huang et al. performed at different temperatures (223 K and 296 K) and precursor
concentration (α-pinene 0.714 and 2.2 ppm) suggested that the physicochemical properties, such as the composition
of the oligomers (at the nanometer scale), can be affected by a variation of temperature (Huang et al., 2018). The broad
range of chemical molecules formed and the impact of the experimental conditions on their character remains a subject
for atmospheric chemistry as well as for combustion chemistry studies. Whatever the mechanism of aerosols
formation, i.e., oligomerization, functionalization, or accretion, their composition will be linked to that of the initial
radical pool (Camredon et al., 2010;Meusinger et al., 2017;Tomaz et al., 2021).
In low-temperature combustion, when the temperature is increased, fuel's autoxidation rate goes through a maximum
between 500 and 670 K, depending on the nature of the fuel (Belhadj et al., 2020;Belhadj et al., 2021c). In low-
temperature combustion chemistry as in atmospheric chemistry, the oxidation of a chemical compound leads to the
formation of several thousands of chemical products which result from successive additions of oxygen, isomerization,
accretion, fragmentation, and oligomerization (Benoit et al., 2021;Belhadj et al., 2021b). The exhaustive analysis of
chemical species remains, under the current instrumental limitations, impossible. Indeed, this would consist in
analyzing several thousands of molecules using separative techniques such as ultra-high-pressure liquid
chromatography (UHPLC) or ion mobility spectrometry (IMS) (Krechmer et al., 2016;Kristensen et al., 2016).
Nevertheless, it is possible to classify these molecular species, considering only $C_xH_yO_z$ compounds, according to
criteria accessible via graphic tools representation such as van Krevelen diagrams, double bond equivalent number
(DBE), and average carbon oxidation state (OSc) versus the number of carbon atoms (Kourtchev et al., 2015;Nozière
et al., 2015). Such postprocessing of large datasets has the advantage of immediately highlighting classes of
compounds or physicochemical properties such as the condensation of molecules (vapor pressure), the large variety
of oxidized products ($C_xH_yO_{1\ to\ 15}$ in the present experiments) and the formation of oligomers (Kroll et al., 2011;Xie et
al., 2020).
In addition to the recent studies focusing on the first steps of autoxidation, a more global approach, based on the
comparison of possible chemical transformations related to autoxidation in low temperature combustion and
atmospheric chemistry, is needed for evaluating the importance of autoxidation under tropospheric and low-
temperature combustion conditions. In order to study the effects of experimental conditions on the diversity of
chemical molecules formed by autoxidation, we have selected α-pinene and limonene, two isomeric terpenes among
the most abundant in the troposphere (Zhang et al., 2018). Limonene has a single ring structure and two double bonds,
one of which is exocyclic. α-Pinene has a bicyclic structure and a single endo-cyclic double bond. These two isomers
with their distinctive physicochemical characters are good candidates for studying autoxidation versus initial chemical
structure and temperature. For α-pinene, in addition to the reactivity of its endo-cyclic double bond, products of ring
opening of the cyclobutyl group have been detected (Kurtén et al., 2015;Iyer et al., 2021), which could explain the
diversity of observed oxidation products. This large pool of oxidation products is increased in the case of limonene by
the presence of two double bonds (Hammes et al., 2019;Jokinen et al., 2015).
The present work extends that concerning the oxidation of limonene alone (Benoit et al., 2021). Compared to previous
works, we have added the study of α-pinene oxidation to that of limonene and investigated the impact of ionization
modes on the number of molecules detected and their chemical nature (unsaturation, oxidation rate). The size of the
experimental and bibliographic databases has been increased by more than 50%, in particular by adding data specific
to autoxidation (Krechmer et al., 2016;Tomaz et al., 2021) and references on α-pinene (Tab. 2)). Here, we oxidized
α-pinene and limonene in a jet-stirred reactor at atmospheric pressure, excess of oxygen, and elevated temperature.
We characterized the impact of using different ionization techniques (HESI and APCI) in positive and negative modes
on the pool of detected chemical formulas. The particularities of each ionization mode were analyzed to identify the
most suitable ionization technique for exploring the formation of autoxidation products under low temperature
combustion. H/D exchange and 2,4-dinitrophenyl hydrazine derivatization were used to assess the presence of
hydroperoxy and carbonyl groups, respectively. Chemical formulas detected here and in atmospheric chemistry studies
were compiled and tentatively used to evaluate the importance of autoxidation routes under both conditions.
**2 Experiments**
**2.1 Oxidation experiments**
The present experiments were carried out in a fused silica jet-stirred reactor (JSR) setup presented earlier (Dagaut et
al., 1986;Dagaut et al., 1988) and used in previous studies (Dagaut et al., 1987;Benoit et al., 2021;Belhadj et al.,
2021c). We studied separately the oxidation of the two isomers, α-pinene and limonene. As in earlier works (Benoit
et al., 2021;Belhadj et al., 2021c), α-pinene (+), 98% pure from Sigma Aldrich and limonene (R)-(+), >97% pure from
Sigma Aldrich, were pumped by an HPLC pump (Shimadzu LC10 AD VP) with an online degasser (Shimadzu DGU-
20 A3) and sent to a vaporizer assembly where it was diluted by a nitrogen flow. Each terpene isomer and oxygen,
both diluted by $N_2$, were sent separately to a 42 mL JSR to avoid oxidation before reaching 4 injectors (nozzles of 1
mm I.D.) providing stirring. The flow rates of nitrogen and oxygen were controlled by mass flow meters. Good thermal
homogeneity along the vertical axis of the JSR was recorded (gradients of < 1 K/cm) by thermocouple measurements
(0.1 mm Pt-Pt/Rh-10% wires located inside a thin-wall silica tube). In order to observe the oxidation of these isomers,
which are not prone to strong self-ignition, the oxidation of 1% of these chemical compounds ($C_{10}H_{16}$) under fuel-lean
conditions (equivalence ratio 0.25, 56% $O_2$, 43% $N_2$), was carried out at 590 K, atmospheric pressure, and a residence
time of 2 s. Under these conditions, the oxidation of the two isomers is initiated by slow H-atom abstraction by
molecular oxygen ($RH + O_2 \rightarrow R^{\bullet} + HO_2^{\bullet}$). The fuel radicals $R^{\bullet}$ react rapidly with $O_2$ to form peroxy radicals which
undergo further oxidation, characteristic of autoxidation. Nevertheless, this autoxidation mechanism, although
predominant, is not exclusive and other oxidation mechanisms are possible (Belhadj et al., 2021b). In this case, there
may be a random overlap of chemical formulas. The autoxidation criteria (two chemical formulas separated by two
oxygen atoms) allows to limit or avoid these overlaps.
**2.2 Chemical analyses**
A 2 mm I.D. probe was used to collect samples. To measure low-temperature oxidation products ranging from early
oxidation steps to highly oxidized products, the samples were bubbled into cooled acetonitrile (UHPLC grade ≥99.9,
T= 0°C, 250 mL) for 90 min. The resulting solution was stored in a freezer at -15°C. The stability of the products was
verified. No detectable changes in the mass spectra were observed after more than one month which is consistent with
previous findings (Belhadj et al., 2021c).
Analyses of samples collected in acetonitrile (ACN) were carried out via direct infusion (rate: 3μL/min and recorded
for 1 min for data averaging) in the ionization chamber of a high-resolution mass spectrometer (Thermo Scientific
Orbitrap® Q-Exactive, mass resolution 140,000 and mass accuracy <0.5 ppm RMS). UHPLC conditions were: a
Vanquish UHPLC Thermo Fisher Scientific with a C18 column (Phenomenex Luna, 1.6 μm, 110 Å, 100x2.1 mm).
The column temperature was maintained at 40°C. 3μml of sample were eluted by a mobile phase containing water-
ACN mix (pure water, ACN HPLC grade) at a flow rate of 250 μL/min (gradient: 5% to 20% ACN -3 min, 20% to
65% ACN - 22 min, 65% to 75% ACN – 4 min, 75% to 90% ACN - 4 min, for a total of 33 min).
Both heated electrospray ionization (HESI) and atmospheric chemical ionization (APCI) were used in positive and
negative modes for the ionization of products. HESI settings were: spray voltage 3.8 kV, vaporizer temperature of
150°C, capillary temperature 200°C, sheath gas flow of 8 arbitrary units (a.u.), auxiliary gas flow of 1 a.u., sweep gas
flow of 0 a.u.. In APCI, settings were: corona discharge current of 3μA, spray voltage 3.8 kV, vaporizer temperature
of 150°C, capillary temperature of 200°C, sheath gas flow of 8 a.u., auxiliary gas flow of 1 a.u., sweep gas flow of 0
a.u.. In order to avoid transmission and detection effects of ions depending on their mass inside the C-Trap (Hecht et
al., 2019), acquisitions with three mass ranges were performed (m/z 50-750; m/z 150-750; m/z 300-750). The upper
limit of m/z 750 was chosen because of the absence of a signal beyond this value. It was shown that no significant
oxidation occurred in the HESI and APCI ion sources by injecting a limonene-ACN mixture (Fig. S1). The
optimization of the Orbitrap ionization parameters in HESI and APCI did not show any clustering phenomenon for
these two monoterpene isomers. The parameters evaluated were: injection source - capillary distance, vaporization
and capillary temperatures, applied difference of potential, injected volume, flow rate of nitrogen in the ionization
source. Positive and negative HESI mass calibrations were performed using Pierce$^{TM}$ calibration mixtures (Thermo
Scientific). Chemical compounds with relative intensity less than 1 ppm to the highest MS signal in the mass spectrum
were not considered. Nevertheless, it should be considered that some of the molecules presented in this study could
result from our experimental conditions (continuous flow reactor, reagent concentration, temperature, reaction time)
and to some extent from our acquisition conditions, different from those in the previous studies (Deng et al.,
2021;Quéléver et al., 2019;Meusinger et al., 2017;Krechmer et al., 2016;Tomaz et al., 2021;Fang et al.,
2017;Witkowski and Gierczak, 2017;Jokinen et al., 2015;Nørgaard et al., 2013;Bateman et al., 2009;Walser et al.,
2008;Warscheid and Hoffmann, 2001;Hammes et al., 2019;Kundu et al., 2012). Operating with a continuous flow
reactor at elevated temperature and high initial concentration of reagents allows the formation of combustion-relevant
products, which does not exclude their possible formation under atmospheric conditions. To assess the formation of
products containing OOH and C=O groups, as in previous works (Belhadj et al., 2021a;Belhadj et al., 2021b), H/D
exchange with $D_2O$ and 2,4-dinitrophenyl hydrazine derivation were used, respectively.
**3 Data Processing**
High resolution mass spectrometry (HR-MS) generates large datasets which are difficult to fully analyze by sequential
methods. When the study requires the processing of several thousands of molecules, the use of statistical tools and
graphical representation means becomes necessary. In this study, we have chosen to use the van Krevelen diagram
(Van Krevelen, 1950) by adding an additional dimension, the double bond equivalent (DBE). The DBE number
represents the sum of unsaturation and rings present in a chemical compound (Melendez-Perez et al., 2016). The
interest of this type of representation is to be able to identify more easily the clusters (increase of the DBE number at
constant O/C and H/C ratios)
$$DBE = 1 + C - H/2$$
This number is independent of the number of O-atoms, but changes with the number of hydrogen atoms. Decimal
values of this number, which correspond to an odd number of hydrogen atoms, were not considered in this study. Then,
the superpositions of points (and therefore of chemical formulas) in the O/C vs. H/C space are suppressed. The
oxidation state of carbon (OSc) provides a measure of the degree of oxidation of chemical compounds (Kroll et al.,
2011). This provides a framework for describing the chemistry of organic species. It is defined by the following
equation:
$$OSc \approx 2\,O/C - H/C$$
**4 Results and discussion**
We studied the oxidation of α-pinene and limonene ($C_{10}H_{16}$) at 590 K, under atmospheric pressure, with a residence
time of 2 s, and a fuel concentration of 1%. Under these conditions, the formation of peroxides by autoxidation at low
temperature should be efficient (Belhadj et al., 2021c), even though the conversion of the fuels remains moderate.

**4.1 Characterization of ionization sources**

First, we have studied the impact of APCI and HESI sources, in positive and negative modes, on the chemical formulas detected. The HESI and APCI sources in positive and negative mode were used and their operating parameters were varied, i.e., temperature, gas flow and accelerating voltage (see Section 2). For each polarity, only ions composed of carbon, hydrogen (even numbers) and oxygen were considered. Molecular duplicates inherent to mass range overlaps were excluded. By following these rules, we obtained a different number of ions depending on the ionization source and the polarity used. Table 1 shows the number of ions according to the experimental conditions and the discrimination rules.

**Table 1.** Number of ions detected for each source in positive and negative modes (by protonation or deprotonation, respectively)

| Ionization source | α-Pinene | | Limonene | |
|---|---|---|---|---|
| APCI | 646 $(R+H)^+$ | 503 $(R–H)^-$ | 1321 $(R+H)^+$ | 1346 $(R–H)^-$ |
| HESI | 594 $(R+H)^+$ | 693 $(R–H)^-$ | 1017 $(R+H)^+$ | 1864 $(R–H)^-$ |

Each combination of ionization sources and polarity generated a set of chemical formulas. To make a meaningful comparison between the positive and negative ions data, the chemical formulas used were the precursors of the ions identified in the mass spectra. These sets have common data, but also specific chemical formulas. For a given ionization source, ~ 50% of the chemical formulas are observed whatever the ionization polarity, i.e., using both polarities one can capture between 30-50% more molecular species (since some of them are ionized under a single mode (+ or –) depending on their chemical structure). Utilizing both ionization polarities is helpful for identifying a larger quantity of species. The HESI source data were compared to the APCI data (Supplement, Tables S1 and S2), showing an increased number (20 to 30%) of chemical formulas detected by HESI. This increase is characterized by a better detection of negatively ionized species and those with a higher DBE. In order to evaluate further the interest for using these ionization sources, we compiled these data in Venn diagrams and proposed a visualization of these sets with a van Krevelen representation; we added the number of DBE in the third dimension (Supplement, Tables S1 and S2).

In positive ionization mode, independently of the ionization source and in addition to the common molecular formulas, we detected products with an O/C ratio < 0.2 whereas in the negative ionization mode, we detected molecular formulas with an O/C ratio > 0.5. In addition to these observations, we noted that HESI is more appropriate for studying products with a large number of unsaturation (DBE > 5), which is probably related to the increase in the number of hydroperoxide and carboxyl groups along with the fact that a heated ionization source favors vaporization of low volatility compounds. Finally, for an optimal detection of the oxidation products, it is necessary to consider the transmission limits of the C-Trap. Here, we could increase by more than 60% the number of molecular formulas detected using several mass ranges for data acquisition (section 2.2). The most appropriate ionization polarity to be used is tied to chemical functions present in products to be detected. We could increase by 30 to 100% the number of chemical formulas detected by using both positive and negative ionization modes. Using HESI is consistent with previous findings indicating ESI is well suited for the ionization of acidic, polar, and heteroatom-containing chemicals (Kekäläinen et al., 2013). To illustrate the present results, HESI (–)-MS spectra are provided in the Supplement (Fig.

S2). The list of all chemical formulas found in limonene and α-pinene samples (HESI negative and positive mode) is
given in the data-supplement file

**4.2 Autoxidation products detected in a JSR**
In order to compare the oxidation of α-pinene and limonene, we compiled the positive and negative ionization data
obtained with APCI (Table S1) and HESI (Table S2) ionization sources to obtain a more exhaustive database. For the
APCI and HESI sources, we distinguished three datasets, two of which are specific to the oxidation of α-pinene and
limonene and one which is common to both isomers. In the following text, "only" will be used to describe the
molecules specific to the oxidation of one of the isomeric terpenes. This common dataset represents more than 90%
of the chemical formulas identified in the α-pinene oxidation samples detected with APCI or HESI. For limonene, for
which the number of identified chemical formulas is larger, regardless of the ionization source, this common data set
represents nearly 40% of the chemical formulas detected. In these two cases, the relatively low residence time (2
seconds) and the diversity of the chemical formulas obtained suggest that the oxidation of these two terpene isomers
leads to ring opening, a phenomenon also observed in atmospheric chemistry (Berndt et al., 2016;Zhao et al., 2018;Iyer
et al., 2021). Concerning the molecular formulas of products common to both isomers, Figure 1 shows that they are
limited to compounds with 10 oxygen atoms or lower. This limit is linked to α-pinene whose oxidation beyond 10
oxygen atoms remains weak (less than 2% of the detected molecules for this terpene). In the case of limonene, the
presence of an exocyclic double bond will increase, in a similar way to atmospheric chemistry (Kundu et al., 2012),
the possibilities of oxidation and accretion. It remains however impossible, considering the size of the whole dataset
and the diversity of the isomers, to formalize all the reaction mechanisms. Nevertheless, the formation of oxidized
species can be described with the help of graphical tools. The number of oxygen atoms per molecule indicates that
limonene oxidizes more than α-pinene (Fig. 1a). In the case of limonene, with a HESI source, chemicals with an
oxygen number of up to 15 were detected. Most of the chemical formulas recorded had 8-10 O-atoms (Fig. 1c), whereas
for α-pinene the products with >8 O-atoms were much less abundant (Fig. 1b). Moreover, for the products specific of
limonene oxidation, this graph shows a distribution centered on 9 oxygen atoms with carbon skeletons probably
resulting from accretion.






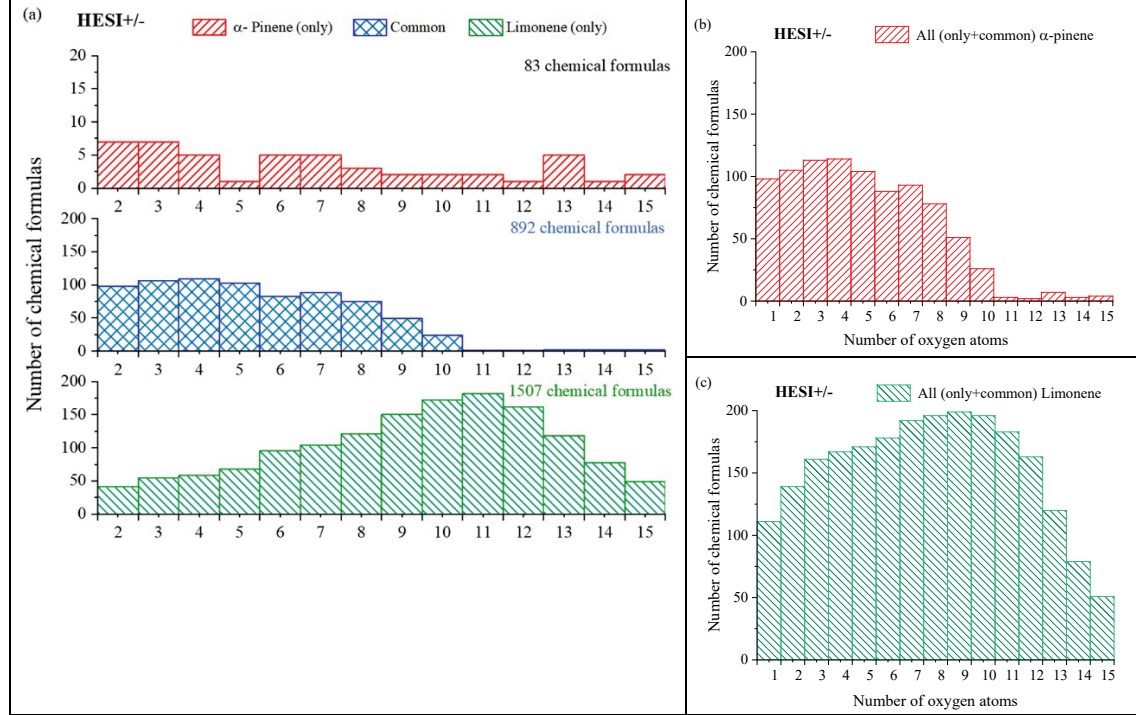

**Figure 1.** Distribution of α-pinene and limonene oxidation products as a function of their oxygen content (ionization source: HESI, combined positive and negative modes data). (a) α-pinene and limonene HESI(+/−), (b) α-pinene HESI(+/−), (c) limonene HESI(+/−)

To verify this accretion hypothesis, we can plot the OSc as a function of the number of carbon atoms or the O/C ratio at fixed number of C-atoms (Fig. 2). Indeed, the presence of chemical compounds with 11 carbon atoms can be explained by an accretion phenomenon (Wang et al., 2021), but the advantage of this OSc vs. nC space representation (Kroll et al., 2011) is to allow studying this phenomenon on all the data. One can visualize the evolution of the molecular oxidation for each carbon skeleton and the formation of oligomers. Species that are unique to one of the isomers, or common to both are differentiated using different colors. In addition, in Fig. 2a, we observe mechanisms of fragmentation (C$_{<10}$), accretion and oligomerization (C$_{>10}$). These reaction mechanisms contribute to forming chemical classes according to their number of carbon atoms, up to C=30. This limit is probably due to the ionization or detection capacity of the spectrometer. The increase in the number of oxygen atoms, but also of carbon atoms will decrease products volatility. Following a classification proposed in the literature (Kroll et al., 2011), we distinguished four sets of products: low volatile oxidized organic aerosols (LV-OOA), semi-volatile oxidized organic aerosols (SV-OOA), biomass burning organic aerosols (BBOA) and water-soluble organic carbons (WSOC). In the OSc versus carbon number plot (Fig 2a), the vertical lines (at constant carbon number) are a first criterion for finding potential candidate products of autoxidation. Figure 2b shows, for a fixed number of carbon and hydrogen atoms, the diversity of oxidized species formed. Different oxygen parities are observed, showing that different reaction mechanisms occur.

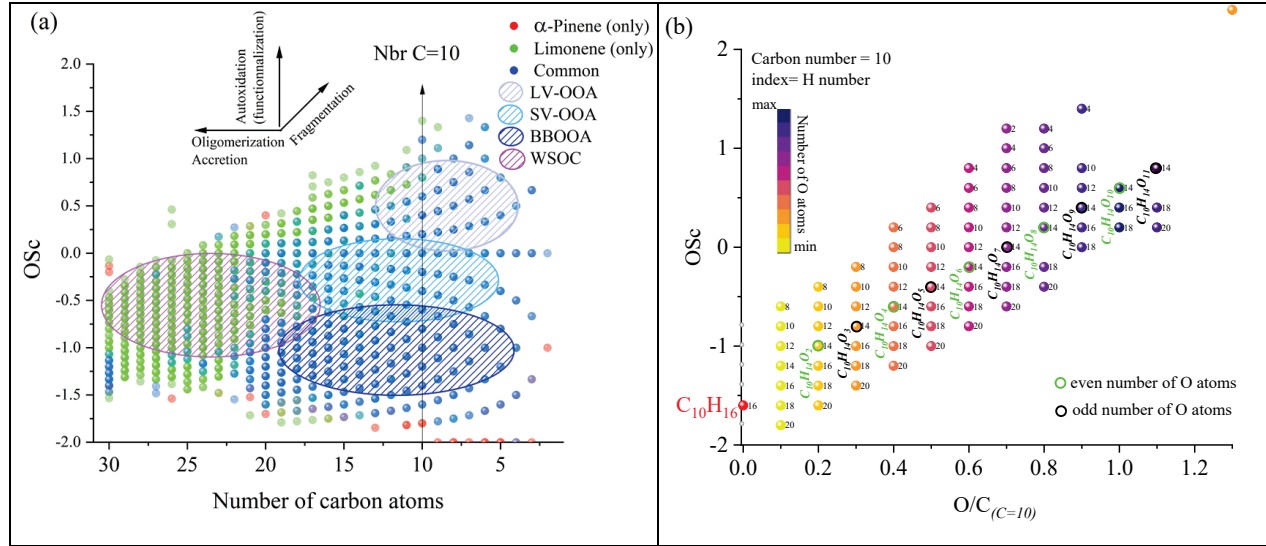

**Figure 2.** Overview of the distribution of limonene and α-pinene oxidation products observed in a JSR: (a) OSc versus carbon number in detected chemical formulas from APCI and HESI data. (b) Molecular formulas detected in this study presented in the OSc versus O/C space for a carbon number of 10; index of the products: number of hydrogen atoms.

This parity distinction is initially present for the two main radicals, ROO˙ and RO˙ involved in autoxidation mechanisms. However, termination and propagation reactions will change the oxygen parity (Fig. 3). Then, parity links between radicals and molecules are lost, which prevents interpretation of radical oxidation routes. HESI data showed an equivalent distribution of oxygen parities in molecular products (odd: 51%, even 49%) which cannot allow concluding on the relative importance of reaction pathways. It should be noted that other reaction mechanisms can also change oxygen parity, e.g., QOOH → cyclic ether (QO) + OH (Wang et al., 2017). Figure 2 (b) illustrates the autoxidation products presented in Fig. S3. There, one can see the formation routes to even-oxygen compounds $C_{10}H_{14}O_{2\ to\ 10}$ and to odd-oxygen compounds $C_{10}H_{14}O_{3\ to\ 11}$. The molecular formulas detected in our experiments as shown in bold in Fig. 2b. The others formulas presented in Fig. 2b should result from others oxidation pathways. Indeed, products with chemical formulas with H≥16 cannot derive from the autoxidation pathways described in Fig S3. Other pathways (Fig. S5) can yield such species, e.g., through the initial addition of OH on terpenes double bonds followed by $O_2$ addition and autoxidation of resulting products.

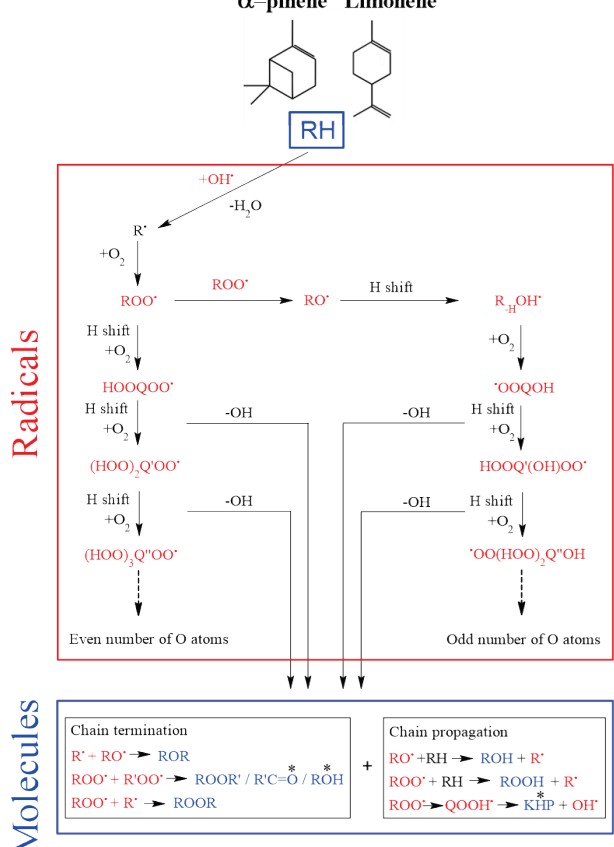

**Figure 3.** Accepted autoxidation reaction mechanisms in combustion (left) and in the atmosphere (left and right). * indicates a change of oxygen atoms parity (Berndt et al., 2016).

Nevertheless, despite this change in parity, in the case of autoxidation, the free-radical reaction pathway (shown in Supplements Figs. S3 and S5 for both oxygen parities) can produce a set of molecular products that mirrors repeated $O_2$ addition, characteristic of autoxidation (Fig. 2b). Furthermore, we studied the relative intensities of identified chemical formulas for alpha pinene and limonene (HESI source). The results presented in Table 2 show overall a decrease in relative intensities of products signal with increasing number of oxygen atoms ($C_{10}H_{14}O_{2,4,6,8...}$; $C_{10}H_{14}O_{3,5,7...}$) for both terpenes. It is clear that the repeated addition of $O_2$ on radicals (Fig S3) associated with the decrease of the relative intensities of the products formed is not sufficient to assess an autoxidation mechanism although it is a necessary step to constrain the identification phase of the isomers, otherwise impossible within sets composed of several thousands of chemical molecules. Finally, few chemical formulas with no chemical relevance (CxH4Oy) were detected. These are probably artefacts linked to the characterization method (ionization mode, ions transfer, ions isolation in the C-trap, incorrect masse identification). We chose to leave these data, knowing that they would be discarded in the various subsequent comparisons.

Table 2: Relative intensities of detected chemical formulas for alpha pinene and limonene (HESI– data) which could result from autoxidation of these terpenes. Signal intensities are given in parentheses. Chemical formulas are highlighted in red in the Supplementary database.

| | Limonene | Alpha-pinene |
|---|---|---|
| **Even number of oxygen atom** | | |
| **hydroxyketone** | $C_{10}H_{14}O_2$ (1.61442E+7) | $C_{10}H_{14}O_2$ (4.54785E+7) |
| **+O$_2$ (1st)** | $C_{10}H_{14}O_4$ (3.34718E+7) | $C_{10}H_{14}O_4$ (2.52885E+6) |
| **+O$_2$ (2nd)** | $C_{10}H_{14}O_6$ (9.58108E+6) | $C_{10}H_{14}O_6$ (7.56393E+4) |
| **+O$_2$ (3rd)** | $C_{10}H_{14}O_8$ (9.55306E+5) | $C_{10}H_{14}O_8$ (2.91182E+4) |
| **+O$_2$ (4th)** | $C_{10}H_{14}O_{10}$ (1.00597E+4) | $C_{10}H_{14}O_{10}$ (not detected) |
| | $C_{10}H_{14}O_{12}$ (not detected) | -- |
| **Odd number of oxygen atom** | | |
| **hydroperoxy carbonyl** | $C_{10}H_{14}O_3$ (3.23297E+7) | $C_{10}H_{14}O_3$ (9.3999E+6) |
| **+O$_2$ (1st)** | $C_{10}H_{14}O_5$ (2.27278E+7) | $C_{10}H_{14}O_5$ (1.17044E+5) |
| **+O$_2$ (2nd)** | $C_{10}H_{14}O_7$ (4.04207E+6) | $C_{10}H_{14}O_7$ (7.17307E+4) |
| **+O$_2$ (3rd)** | $C_{10}H_{14}O_9$ (1.92816E+5) | $C_{10}H_{14}O_9$ (not detected) |
| **+O$_2$ (4th)** | $C_{10}H_{14}O_{11}$ (6.34129E+3) | -- |
| | $C_{10}H_{14}O_{13}$ (not detected) | -- |

## 4.3 Combustion versus atmospheric oxidation

### 4.3.1 Global analysis

We have explored potential chemical pathways related to autoxidation in the previous Section. For this purpose, we have performed experiments under cool flame conditions (590 K). This autoxidation mechanism is also present in atmospheric chemistry, but it is only recently that it has been found that this mechanism could be one of the main formation pathways for SOA (Savee et al., 2015;Crounse et al., 2013;Jokinen et al., 2014a;Iyer et al., 2021). Studies have described this mechanism in the case of atmospheric chemistry with the identification of radicals and molecular species (Tomaz et al., 2021). However, previous studies on the propagation of this reaction mechanism have mainly focused on the initial steps of autoxidation without screening all identified chemical formulas for potential autoxidation products. It is therefore useful to assess the proportion of possible autoxidation products among the total chemical species formed.

Here, we propose a new approach which consists in assessing a set of molecules mainly resulting from autoxidation against different sets of experimental studies related to atmospheric chemistry. The objective is to evaluate similarity of oxidation products formed under these conditions. For this purpose, we selected a HESI ionization source, better suited for detecting higher polarity oxidized molecules, as well as higher molecular weight products (detection of 96% of the total chemical formulas observed in autoxidation by APCI and HESI).

Among published atmospheric chemistry studies of terpenes oxidation, we have selected 15 studies presenting enough chemical products of oxidation, 4 for α-pinene and 11 for limonene. The data were acquired using different experimental procedures (methods of oxidation, techniques of characterization). Table 3 summarizes all the

experimental parameters related to the selected studies. From that Table, one can note that few studies involved
chromatographic analyzes (Tomaz, 2021; Witkowski an Gierczak, 2017; Warscheid and Hoffmann, 2001). The data
are from the articles or files provided in the Supplement Tables S1 and S2. In these studies, oxidation was performed
only by ozonolysis with different experimental conditions that gather the main methods described in the literature:
ozonolysis, dark ozonolysis, ozonolysis with OH scavenger, ozonolysis with or without seed particles. We considered
that the ionization mode used in mass spectrometry did not modify the nature of the chemical species but only the
relative detection of ions, depending on the type of ionization used, and the sensitivity of the instruments (Riva et al.,
2019). The combination of data obtained using (+/–) HESI gives a rather complete picture of the autoxidation products.
First, we compared the data from ozonolysis studies of each terpene and identified similarities through Venn diagrams.
For studies with two ionization sources, duplicate chemical formulas were removed. We selected the four most
representative studies by the number of the chemical formulas detected. Then, we compared the set of chemical
formulas identified after ozonolysis to those produced in low-temperature combustion, the objective being (i) to
highlight similarities in terms of products generated by the two oxidation modes and (ii) to identify chemicals resulting
from autoxidation.


**Table 3.** Experimental settings of 15 oxidation studies of two terpenes under atmospheric conditions and cool flames
(LC stands for liquid chromatography).

| Reference | Oxidation mode | Sampling | Experimental setup | Concentrations of reactants | Ionization /source | Instrument | Chemical formulas | LC |
|---|---|---|---|---|---|---|---|---|
| | | | | α-Pinene | | | | |
| Y. Deng et al. (2021) | Dark ozonolysis seed particles OH scavenger | online | Teflon bag; 0.7m³ | 3.3±0.6 ncps ppbv⁻¹ α -Pinene | ESI | ToF-MS | 351 | No |
| Quéléver et al. (2019) | Ozonolysis | online | Teflon bag 5 m³ | 10 & 50 ppb α -Pinene | $NO_3^-$ (CI) | CI-APi-TOF | 68 | No |
| Meusinger et al. (2017) | Dark Ozonolysis OH scavenger no seed particles | offline | Teflon bag 4.5 m³ | 60 ppb α -Pinene | Proton transfer | PTR-MS-ToF | 153 | No |
| Krechmer et al. (2016) | Ozonolysis | offline | PAM Oxidation reactor | Field measurement | ESI (−) and $NO_3^-$ (CI) | CI-IMS-ToF | 43 | No |
| This work | Cool-flame autoxidation | offline | Jet-stirred reactor 42 ml | 1%, α -pinene No ozone | APCI(3kV) HESI (3kV) | Orbitrap® Q-Exactive | 820 (APCI) 975 (HESI) | Yes |
| | | | | Limonene | | | | |
| Krechmer et al. (2016) | Ozonolysis | offline | PAM Oxidation reactor | not specified | ESI (−) and $NO_3^-$ (CI) | CI-IMS-ToF | 63 | No |
| Tomaz et al. (2021) | Ozonolysis | online | Flow tube reactor (18L) | 45-227 ppb limonene | $NO_3^-$ (CI) - Neg | Orbitrap® Q-Exactive | 199 | Yes |
| Fang et al. (2017) | OH-initiated photooxidation dark ozonolysis | online | Smog chamber | 900–1500 ppb limonene | UV; 10 eV | Time-of-Flight (ToF) | 17 | No |
| Witkowski and Gierczak (2017) | Dark ozonolysis | offline | Flow reactor | 2 ppm, limonene | ESI,4.5 kV | Triple quadrupole | 12 | Yes |
| (Jokinen et al., 2015) | Ozonolysis | online | Flow glass tube | 1–10000 x10⁹ molec.cm⁻³, limonene | $NO_3^-$ (CI) | Time-of-Flight (ToF) | 11 | No |
| Nørgaard et al. (2013) | Ozone (plasma) | online | direct on the support | 850 ppb ozone 15-150 ppb limonene | plasma | Quadrupole time-of-flight (QToF) | 29 | No |
| Bateman et al. (2009) | Dark and UV radiations ozonolysis | offline | Teflon FEP reaction chamber | 1 ppm ozone 1 ppm limonene | modified ESI (+/−) | LTQ-Orbitrap Hybrid Mass (ESI) | 924 | No |
| Walser et al. (2008) | Dark ozonolysis | offline | Teflon FEP reaction chamber | 1-10 ppm ozone 10 ppm limonene | ESI (+/−); 4.5 kV | LTQ-Orbitrap Hybrid Mass (ESI) | 465 | No |
| Warscheid & Hoffmann (2001) | Ozonolysis | online | Smog chamber | 300-500 ppb limonene | APCI; 3kV | Quadrupole ion trap mass | 21 | Yes |
| Hammes et al., (2019) | Dark ozonolysis | online | Flow reactor | 15, 40, 150 ppb limonene | ²¹⁰ Po α acetate ions | HR-ToF-CIMS | 20 | No |
| Kundu et al. (2012) | Dark ozonolysis | offline | Teflon reaction chamber | 250 ppb ozone 500 ppb limonene | ESI; 3.7 and 4 kV | LTQ FT Ultra, Thermo Sct (ESI) | 1197 | No |
| This work | Cool-flame autoxidation | offline | Jet-stirred reactor 42 ml | 1%, limonene No ozone | APCI(3µA) HESI (3kV) | Orbitrap® Q-Exactive | 1863(APCI) 2399(HESI) | Yes |


For α-pinene oxidation, in the four selected studies 567 chemical formulas were detected, all polarities combined.
Only one study (Meusinger et al., 2017) was performed in positive ionization mode and none of the studies reported
data were obtained with two ionization modes (+/–). For the oxidation of limonene, the four selected studies identified
1434 chemical formulas. Among these studies, the experiments by Walser et al. were performed with both (+) and (–
) ionization modes. In contrast to the α-pinene case, the selected studies for limonene were performed with similar
ionization sources, which probably contributed to increased data similarity (Walser et al., 2008). In the case of
limonene oxidation, for which accretion is more important than for α-pinene, and for which a greater number of
chemical formulas were identified, the similarities are more important (Jokinen et al., 2014b). These results are
presented in Figure 4 where the ionization polarity used in each study is specified.

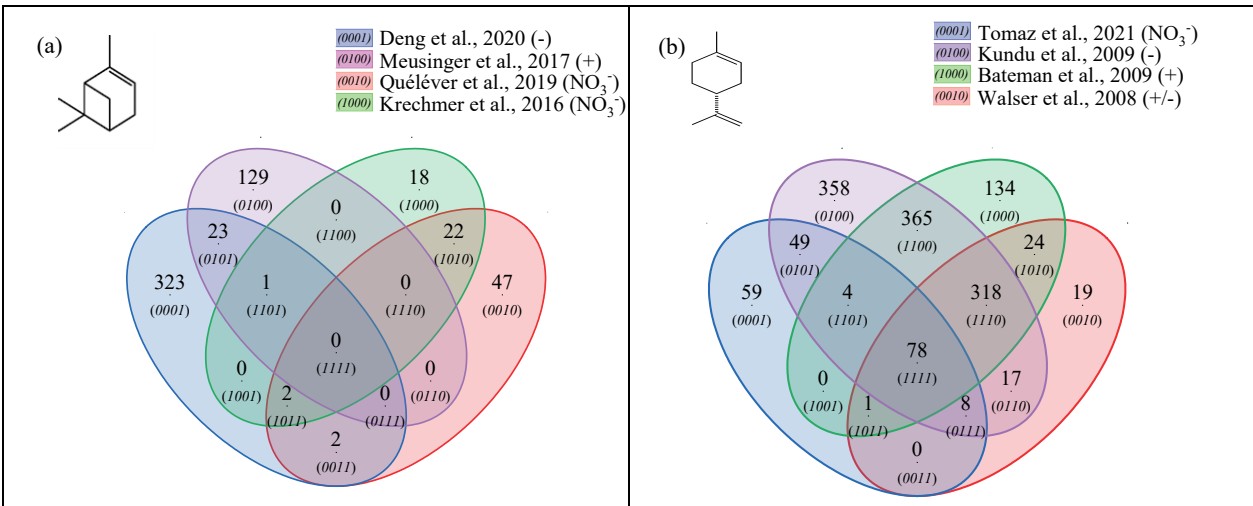

**Figure 4.** Venn diagrams for comparing the oxidation results from ozonolysis of (a) α-pinene and (b) limonene (see conditions in Table 1). Each digit indicates a study, the value of the digit characterizes the presence (value 1) or absence (0) common products detected in different studies, e.g., 23 chemical formulas (0101) (Fig. 4a) are common to the studies of Deng et al; (0001) and Meusinger et al. (0100)

For α-pinene, no chemical formula is common to all datasets. Different hypotheses can be offered to explain this
result. Among them, the number of chemical formulas identified per study remains limited (a few dozen to several
hundred) and these small datasets are sometimes restricted to specific mass ranges, e.g. $C_{10}$ to $C_{20}$ (Quéléver et al.,
2019). In the case of studies carried out with an $NO_3^-$ source, sensitive to HOMS, produced preferentially by
autoxidation, we note that nearly 50% of the chemical formulas (10/22; (*1010*)) are linked by a simple difference of 2
oxygen atoms.
For limonene, chemical formulas are common to the four studies selected here. In this data set, a large majority of
chemical formulas show a similar relationship to autoxidation, i.e., a simple difference of two oxygen atoms: 62%
(Tomaz et al., 2021), 54% (Walser et al., 2008), 69% (Kundu et al., 2012), 66% (Bateman et al., 2009) and 72% (this
study). This result seems to indicate that autoxidation dominates.
One can then ask if reaction mechanisms common to atmospheric and combustion chemistry can generate, despite of
radically different experimental conditions, a set of common chemical formulas and if in this common dataset, a
common link, characteristic of autoxidation, is observable? To address that question, we compared all the previous
results, for each of these terpenes to those obtained under the present combustion study. The comparisons were made
using our HESI data. One should remember that the oxidation conditions in the JSR were chosen in order to maximize
low-temperature autoxidation. Again, we used Venn diagrams to analyze these datasets consisting of 1590 chemical
formulas in the case of α-pinene and 5184 chemical formulas in the case of limonene. The results of these analyses
are presented in Figure 5.
It turned out that for α-pinene, 301 chemical formulas and for limonene 871 chemical formulas were common to
oxidation by ozonolysis (with or without scavenger) and combustion. This represents 31% of the chemical formulas
for the ozonolysis of α-pinene and 36% for those of limonene ozonolysis. For α-pinene, the similarities compared to
combustion are specific to each study: (Deng et al., 2021) 69% (243), (Meusinger et al., 2017) 46%, (71) (Quéléver et
al., 2019) 7% (5), (Krechmer et al., 2016) 23% (10). Chemical formulas common to all studies were not identified.
This lack of similarity may be due to a partial characterization of the chemical formulas, a weaker oxidation of α-
pinene with an ionization mode less favorable to low molecular weights products.

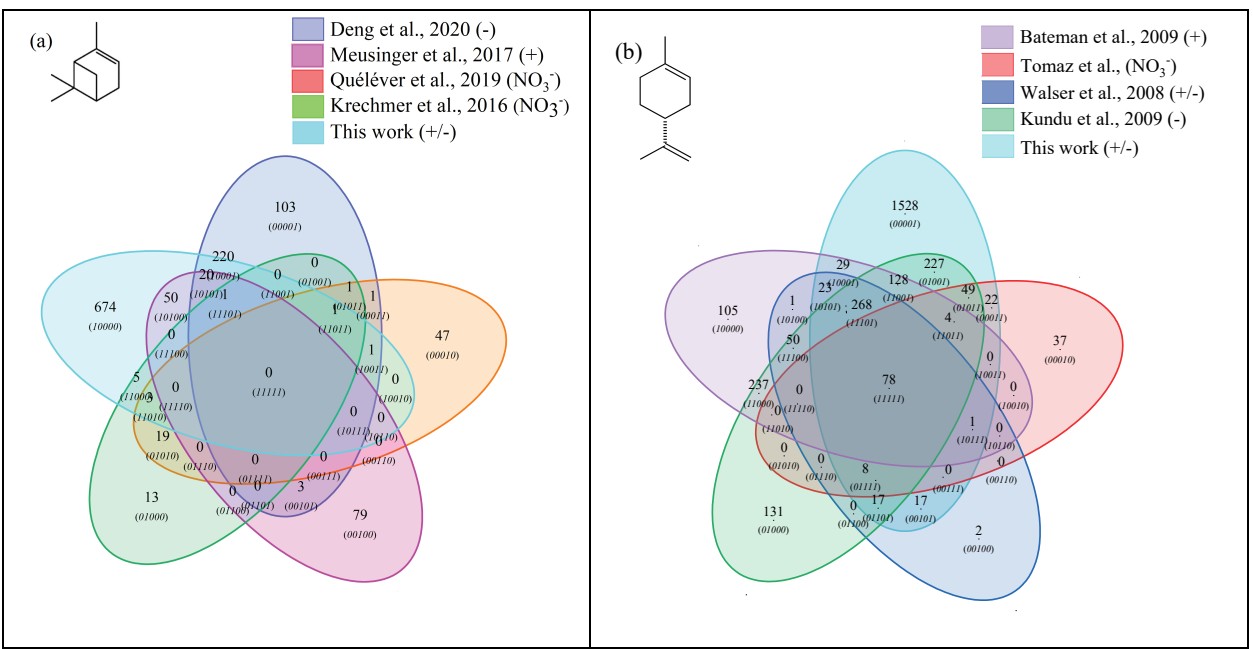

**Figure 5.** Venn diagrams comparing the oxidation results from ozonolysis and combustion of (a) α-pinene and (b)
limonene (see conditions in Table 1).
For limonene, the similarities with combustion are more important and less spread out. They represent for the different
studies: 65% (Kundu et al., 2012), 88% (Walser et al., 2008), 81% (Tomaz et al., 2021), and 57% (Bateman et al.,
2009). Moreover, there is a common dataset of 78 chemical formulas which can derive from autoxidation mechanisms.
Considering the very different experimental conditions, we must wonder about the impact of the double bonds in this
similarity. In the case of limonene, we think their presence will indeed promote the formation of allylic radicals and
then peroxide radicals (one of the motors of autoxidation). It is necessary to specify again that different reaction
mechanisms can cause the observed similarities. However, the preponderance of autoxidation in so-called cool flame
combustion is obvious, and in atmospheric chemistry, this reaction mechanism is competitive or dominates (Crounse
et al., 2013;Jokinen et al., 2014a). If we search for an autoxidation link between these 78 chemical formulas, we
observe that 45% of these chemical formulas meet this condition: difference of two oxygen atoms between formulas,
at constant number of carbon and hydrogen atoms (Data supplements, tab-5) More precisely, these molecules are
centered in a van Krevelen diagram on the ratios O/C=0.6 and H/C=1.6, in the range 0.29 < O/C < 0.77 and 1.33 <
H/C < 1.8. All oxidized molecules associated with this dataset are presented in Figure 6. The dispersion of chemical

formulae, far from being random, could be correlated with an autoxidation mechanism where the number of carbon and hydrogen atoms are constant.

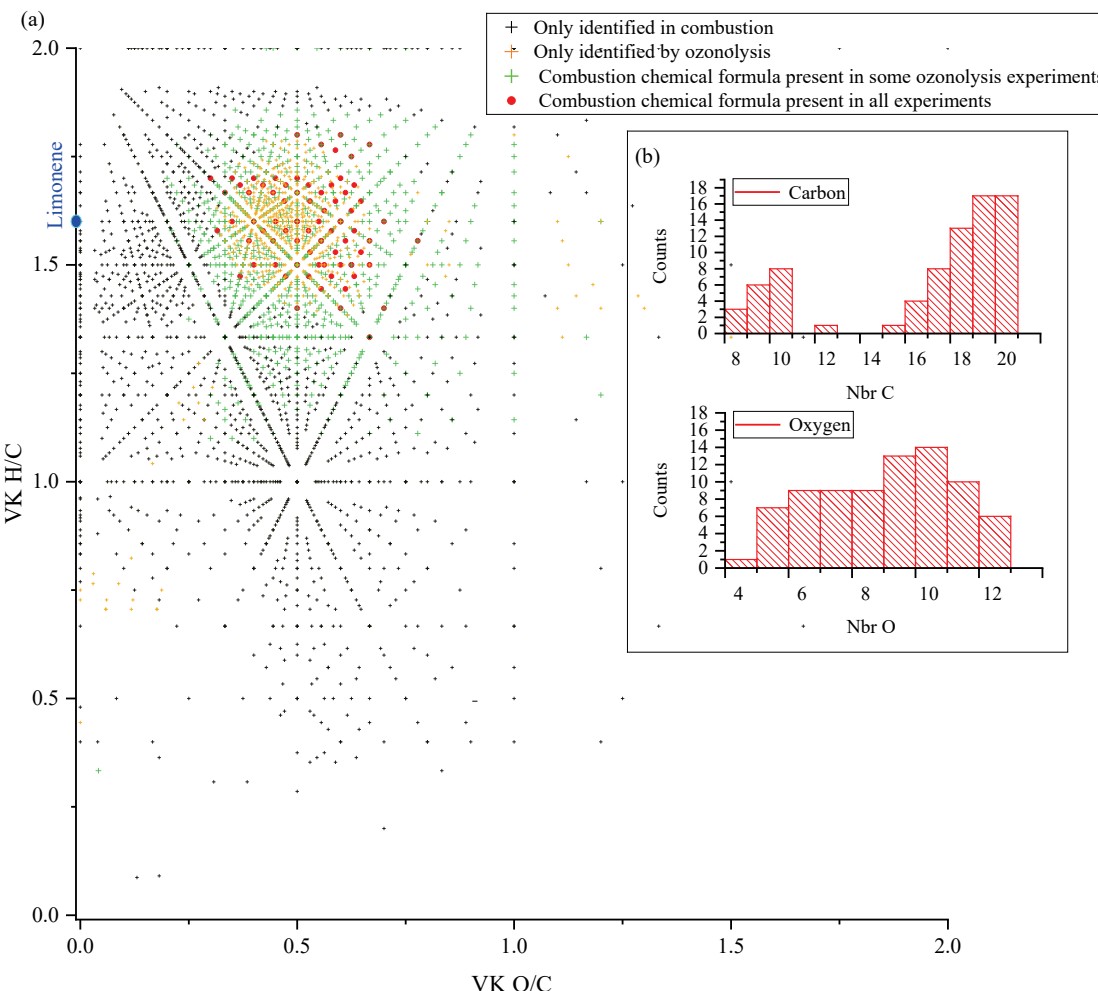

**Figure 6.** (a) Van Krevelen diagram showing specific and common chemical formulas detected after to oxidation of limonene by ozonolysis and combustion; insert (b): distributions of the number of carbon and oxygen atoms in the 78 chemical formulas common to all experiments.

A 3-D representation of all limonene oxidation data is given in Supplement (Fig. S4a) where DBE is used as third dimension. From that figure, one can note that products with higher DBE (DBE>10) are preferably formed under JSR conditions, i.e., at elevated temperature. A 2D representation (OSc vs DBE, Fig. S4b) completes this 3D view. The corresponding chemical formulas with DBE > 10 could correspond to carbonyls and / or cyclic ethers (˙QOOH → carbonyl + alkene + OH and / or cyclic ether + OH˙). Specificities and similarities of these two oxidation modes (ozonolysis/combustion) were further investigated by plotting the distribution of the number of oxygen atoms in detected chemical formulas (Fig. 7). Indeed, the distribution of the number of oxygen atoms allows, in addition to the Van Krevelen diagram, to provide some additional details on these two modes of oxidation. In ozonolysis, we observed the chemical formulas having the largest number of oxygen atoms. There, oxidation proceeds over a long reaction time where the phenomenon of aging appears through accretion or oligomerization. In combustion, the number of oxygen atoms remains limited to 18, with a lower number of detected chemical formulas compared to the case of ozonolysis. JSR-FIA-HRMS data indicated many more sets of chemical formulas differing by two O-atoms in the

range $C_{10}H_xO_{2-10}$ ($C_{10}H_{12}O_{2-10}$, $C_{10}H_{12}O_{3-9}$, $C_{10}H_{14}O_{2-10}$, $C_{10}H_{14}O_{3-9}$, $C_{10}H_{16}O_{2-8}$, $C_{10}H_{16}O_{3-9}$); see Supplementary
database, tab-4. Although at this stage one cannot prove these species were formed through autoxidation, their formulas
are consistent with autoxidation products. It is in combustion that we observed the highest O/C ratios, indicating the
formation of the most oxidized products. This difference, however, does not affect the similarities between the
chemical formulas detected in the two modes of oxidation. Finally, the analysis of the parities in oxygen atoms, very
similar for the three datasets, confirms that the reaction mechanisms presented in Figure 3 do not allow a simple link
to be established between the oxygen parity of radicals and that of the detected molecular products. The list of these
78 chemical formulas is given in the data-supplement file

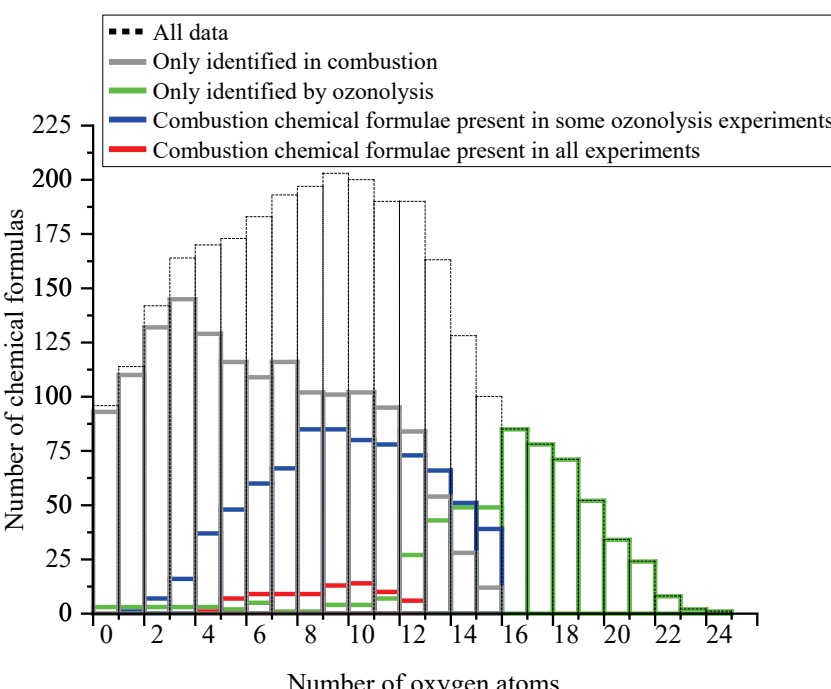

**Figure 7.** Oxygen number distribution for all the molecules identified for the oxidation of limonene: only in
combustion, only in ozonolysis and common to both processes.
*4.3.2 Identification of common isomers.*
We identified a set of chemical formulas common to both atmospheric and combustion oxidation modes and suggested
that this might result from an autoxidation mechanism. We detected several chemical formulas within this dataset that
differ by two oxygen atoms on the same skeleton ($C_{10}H_{16}O_x$). Some of these chemical formulas were previously
identified in Figure 2b or in the 78 common chemical formulas. Focusing on the early stages of limonene oxidation,
there are several chemical formulas, starting from $C_{10}H_{16}O_2$ and $C_{10}H_{16}O_3$, which contain increasing (by 2) number of
oxygen atoms. Table 4 presents the identified chemical formulas with Venn index given in parentheses and ions
intensity. The index for combustion is the rightmost (xxxx1).
Table 4. Products of multiple addition of oxygen on limonene oxidation product by OH. Chemical formulas are
highlighted in red in the Supplementary database.

| First stages of oxidation | 1st addition | 2nd addition | 3rd addition | 4th addition | 5th addition |
|---|---|---|---|---|---|
| $C_{10}H_{16}O_2$ *(10101)* 3.64425E+6 | $C_{10}H_{16}O_4$ *(11101)* 1.01228E+7 | $C_{10}H_{16}O_6$ *(11111)* 1.99061E+6 | $C_{10}H_{16}O_8$ *(01011)* 8.62699E+4 | $C_{10}H_{16}O_{10}$ *(00011)* 4.33184E+4 | $C_{10}H_{16}O_{12}$ *(00010)* |
| $C_{10}H_{16}O_3$ *(11101)* 1.2035E+7 | $C_{10}H_{16}O_5$ *(11111)* 5.91408E+6 | $C_{10}H_{16}O_7$ *(11111)* 4.90565E+5 | $C_{10}H_{16}O_9$ *(01011)* 2.08502E+4 | $C_{10}H_{16}O_{11}$ *(00010)* | |


A proposed formation route of the $C_{10}H_{16}O_x$ species is provided in the supporting (Scheme S5). As can be seen from
there, an autoxidation mechanism can start from $^•C_{10}H_{15}O_4$ and $^•C_{10}H_{15}O_6$, yielding odd-oxygen compounds shown in
Table 4. For even-oxygen compounds, one could propose they are formed after production of RO$^•$, through reaction
R3 (Fig. S3) or decomposition of ROOH to yield RO• and OH•, and also through oxidation of •$C_{10}H_{17}O_5$ (Fig. S5).
Products of additions of two oxygens are also observed for other chemical formulas within this common dataset. To
further investigate the possible formation of common products through atmospheric and combustion chemistries,
UHPLC-HRMS experiments were performed. The chemical compounds $C_{10}H_{16}O_2$ for limonene and $C_{10}H_{16}O_3$ for α-
pinene were selected considering the availability of standards from suppliers and were among the most frequently
reported products in atmospheric chemistry studies (Table 5). Our study shows same retention times for these standards
and isomers detected in combustion samples (Fig S6). This result is more obvious for limonoaldehyde (11.5 min) than
for pinonic acid (3.9 min). In addition, we detected the presence of -OH or -OOH groups by H/D exchange with $D_2O$
and C=O groups through derivatization of carbonyls with 2,4-DNPH for these two chemical formulas (Fig. S7). The
low intensity of H/D exchange for α-pinene oxidation products indicates that the pinonic acid isomer is probably
present at low concentration in the sample. Unfortunately, coelution did not fully allow exploiting MS/MS
fragmentation carried out on the two chemical formulas, and to formally identify the two compounds. There is still a
lot of characterization work to be done, but the hypothesis of common isomeric products formed through an
autoxidation mechanism operating in atmospheric and low-temperature combustion conditions seems to be confirmed.
**Table 5.** Isomers of α-pinene and limonene oxidation reported in the literature.

| | $C_{10}H_{16}O_2$ | | $C_{10}H_{16}O_3$ | |
|---|---|---|---|---|
| α-pinene | Pinonaldehyde | (Fang et al., 2017) | Pinonic acid | (Fang et al., 2017;Ng et al., 2011;Meusinger et al., 2017) |
| | hydroxyketone | (Fang et al., 2017) | hydroxy pinonaldehydes | (Fang et al., 2017;Meusinger et al., 2017) |
| Limonene | limononaldehyde | (Fang et al., 2017;Walser et al., 2008;Bateman et al., 2009) | limononic acid | (Fang et al., 2017;Witkowski and Gierczak, 2017;Hammes et al., 2019;Walser et al., 2008;Bateman et al., 2009;Warscheid and Hoffmann, 2001) |
| | 4-isopropenyl-methylhydroxy-2-oxocyclohexane | (Fang et al., 2017) | 7-hydroxy-limononaldehyde | (Fang et al., 2017;Walser et al., 2008;Bateman et al., 2009;Meusinger et al., 2017) |


**5 Conclusion**
The oxidation of limonene-oxygen-nitrogen and α-pinene-oxygen-nitrogen mixtures was carried out using a jet-stirred
reactor at elevated temperature (590 K), a residence time of 2 s, and atmospheric pressure. The products were analyzed
by liquid chromatography, flow injection, and soft ionization-high resolution mass spectrometry. H/D exchange and
2,4-dinitrophenyl hydrazine derivatization were used to assess the presence of OOH and C=O groups in products,
respectively. We probed the effects of the type of ionization used in mass spectrometry analyses on the detection of
oxidation products. Heated electrospray ionization (HESI +/–) and atmospheric pressure chemical ionization (APCI
+/–) were used. A large dataset was obtained and compared with literature data obtained during the oxidation of
limonene and α-pinene under simulated tropospheric and low-temperature oxidation conditions. This work showed a
surprisingly similar set of chemical formulas of products, including oligomers, formed under the two rather different
conditions, i.e., cool flames and simulated atmospheric oxidation. Data analysis involving van Krevelen diagrams,
oxygen number distribution, oxidation state of carbon, and chemical relationship between molecules, indicated that a
subset of chemical formulas is common to all experiments independently of experimental conditions. More than 35%
of the chemical formulas detected in combustion chemistry experiments using a JSR have been detected in the studies
carried out under atmospheric conditions. Finally, we have outlined the existence of a substantial common dataset of
autoxidation products. This result tends to show that autoxidation is indeed inducing similarity between atmospheric
and combustion products. Detailed analysis of our data was performed by UHPLC-MS/MS of selected chemical
formulas observed in the literature. Nevertheless, final identification was not possible due to coelutions.
The present JSR data could be useful to atmospheric chemists working in the field of wildfire and/or biomass burning
induced air pollution. Considering that low-temperature oxidation (cool flame) products, i.e., VOCs, can be emitted
from biomass burning, wildfires and engine exhausts, the present data should be of interest for the atmospheric
chemists because they complement those obtained in atmospheric chemistry literature. It would be interesting to
complement the atmospheric relevant data with $MS^2$ analyses of products and assessment of the presence of
hydroperoxyl and carbonyl groups HOMs. Further $MS^2$ characterizations are also needed for the products observed in
the present work. Finally, a study of the temperature dependence of products formation would be very useful, both
under cool flame conditions and simulated atmospheric oxidation conditions.

**Acknowledgements**
The authors gratefully acknowledge funding from the Labex Caprysses (ANR-11-LABX-0006-01), the Labex
Voltaire (ANR-10-LABX-100-01), CPER, and EFRD (PROMESTOCK and APPROPOR-e projects) and the French
MESRI for a Ph.D. grant. We also thank (Tomaz et al., 2021) for sharing experimental data on limonene oxidation.

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
