# Peer review of "On the formation of highly oxidized pollutants by autoxidation"

_Atmospheric Chemistry and Physics, 2022_

## Author Response (AR1)

**Dear Reviewer (1),**

Thank you for your comments and suggestions. We have taken them into account in the revised manuscript.

In this document, you will find changes made in response to your comments

Best regards,

Roland BENOIT et al.

*nota bene: In the final version of the corrected paper, some modifications have been made in the light of the comments of the two other reviewers.*

**1/Reviewer 1**

**Comments:** The manuscript entitled "On the formation of highly oxidized pollutants by autoxidation of terpenes under low-temperature combustion conditions: the case of limonene and α-pinene" is the second submission of this work to ACP. The presents an investigation of the cool-flame autoxidation of the two atmospherically-abundant monoterpenes in connection with the formation of highly-oxidized molecules under the atmospherically-relevant conditions.

The author's workload is excellent, and the manuscript is noticeably improved as compared with the previous submission. The authors clearly focused on addressing the previous reviewers' comments and, in connection with the previous criticism, carried out additional experiments. The presentation and the discussion of the results in the context of atmospheric chemistry is also noticeably better.

From the technical point of the article is scientifically sound, and methods and experimental details are presented well. The use of the English language is also significant. At the same time, their writing style is still very unfocused, providing too many experimental details, unnecessary sentences, and repetitions in a number of places. These should be addressed via further corrections.

**Answer**:

We thank the reviewer for his comments and suggestions which we tried to address in the revision

**2/Reviewer 1**

**Comment**: Line 363 Can the authors discuss the possible causes for such behavior, mechanistically wise? Are unsaturated molecules (first-generation products is the oxidation of limonene) more prone to undergo autooxidation?

**Answer**:

The presence of double bonds will indeed promote the formation of allylic radicals and the fixation of oxygens. We can cite this reference "Taking the comparison between n-pentane and 1-pentene, shown in Fig. 26 as an example, it can be seen that the bond dissociation energy for the secondary allylic C–H bond is about 13.0 kcal mol 1 lower than that for the normal secondary C–H bond, which makes the formation of allylic radicals in 1-pentene oxidation very competitive" Chong-WenZhou et al., DOI : 10.1016/j.pecs.2021.100983.We propose this modification line 372**:** Considering the very different experimental conditions, we must wonder about the impact of the double bonds in this similarity. In the case of limonene, we think their presence will indeed promote the formation of allylic radicals and then peroxide radicals (one of the motors of autoxidation).

**3/Reviewer 1**

**Comment:** Section 2.1. The authors seem to be discussing their results under the assumption that no (or almost no) other reaction pathways exist under the experimental conditions used (cool-flame oxidation), aside from autooxidation. Can such an assumption be properly justified? If, for instance, OH is generated in the flame, it will generate OH oxidation products, that will obviously overlap (formulas-wise) with the products generated via dark ozonolysis in the absence of scavengers.

**Answer:**

A clarification has been made line 143: Nevertheless, this autooxidation mechanism, although the predominant one, is not exclusive and other oxidation mechanisms are possible Belhadj et al. 10.3390/molecules26237174. In this case, there may be a random overlap of chemical formulas. The autooxidation criteria (two chemical formulas separated by two oxygen atoms) allows to limit or avoid these overlaps.

**4/Reviewer 1**

**Comment:** 409-410 I would be careful with such definitive statements; the presence of the same elemental composition does not necessarily mean that these are the same molecules, observed in the chamber of flow-tube experiments. The same reasoning should be applied to Table 3

**Answer:**

A modification has been made line 418: Furthermore, we verified that the chemical formulas of the main isomers identified in atmospheric chemistry were present

**5/Reviewer 1**

**Comment:** Table 3. The authors can state that they detected cis-pinonic acid and other compounds listed only after they obtain a sufficient separation of the isomers detected (which wasn't possible) followed by a comparison of the retention time of the standard. This would only be possible for cis-pinonic acid (without resorting to synthesizing the other molecules listed) but from the discussion presented in section 4.3.2 it appears that the authors did not analyze CPA standard.

**Answer:**

We analyzed by UHPLC, with the same experimental conditions of our study, two standards (limononaldehyde and Pinonic acid). See section 4.3.2.

**6/Reviewer 1**

**Comment:** Section 4.3.2. I understand that the motivation behind carrying out LC/MS analyses was to unambiguously prove that the same molecules (?) are formed from combustion-initiated autooxidation and from ozonolysis/OH reaction of α-pinene and limonene? However, this goal was not accomplished, the authors found that compounds with -OH or -OOH and C=O functionalities are formed via the autooxidation mechanism but did not show which specific products present in the chamber and field samples are also formed in their cool-flame oxidation. The LC/MS section, in its current form, does not provide any more information than the above-discussed DI-MS measurements.

**Answer:**

We propose this modification line 420 : We have analyzed by UHPLC, with the same experimental conditions as our study, two standards limononaldehyde and Pinonic acid. These two respective isomers of $C_{10}H_{16}O_2$ and $C_{10}H_{16}O_3$ are among the most studied isomers in atmospheric chemistry. Our study shows a superposition of the elution times of these standards with the isomers detected in combustion

(Fig S4). This superposition is more evident for limonoaldehyde (11.5 min) than for acid Pinonic at 3.9 min. Unfortunately, coelution did not fully allow exploiting MS/MS fragmentation carried out on the two chemical formulas, and to formally identify the two compounds. (

**7/Reviewer 1**

**Comment**: Technical comments: 74-75 Awkwardly phased

**Answer**:

We propose this modification line 74: To better understand the importance of these reaction pathways, the experimental conditions unique to these two chemistries must be considered.

**8/Reviewer 1**

**Comment**: 148-149 Please clarify what was measured with HPLC, it is also customary to provide more column parameters (length, ID, and particle size as well as the manufacturer) and a more detailed gradient elution program – 5% held for x min then linear gradient too.). Was eluent A pure water or did it contained any additive to adjust the pH? What instrument (liquid chromatograph) was used?

**Answer**:

We propose this modification line 152: UHPLC conditions were: A Vanquish UHPLC Thermo Fisher Scientific with a C18 column (Phenomenex Luna, 1.6□□m, 110 Å, 100x2.1 mm). The column temperature was maintained at 40°C. 3□ml of sample were eluted by mobile phase containing water-ACN mix (pure water, ACN HPLC grade) at a flow rate of 250 μL/min (gradient: 5% to 20% ACN -3 min, 20% to 65% ACN - 22 min, 65% to 75% ACN – 4 min, 75% to 90% ACN - 4 min, for a total of 33 min)

**9/Reviewer 1**

**Comment**: 151 "Direct infusion"; **Answer**: done

**Comment**: 203-204 Repeated information from lines 167-168; **Answer**: line 209-210 removed

**10/Reviewer 1**

**Comment**: Line 274 participating in autooxidation?

**Answer**:

We propose this modification line 280: This parity distinction is initially present for the two main radicals, ROO˙ and RO˙, involved in autooxidation mechanisms.

**11/Reviewer 1**

**Comment**: Line 355 In the absence of scavengers, ozonolysis also involves a reaction with the OH. Some authors use the term "ozone-initiated oxidation".

**Answer**:

We propose this modification line 361: It turned out that for α-pinene, 301 chemical formulas and for limonene 871 chemical formulas were common to oxidation by ozonolysis (with or without scavenger) and combustion

**Dear Reviewer (2),**

Thank you for your comments and suggestions. We have taken them into account in the revised manuscript.

In this document, you will find changes made in response to your comments

Best regards,

Roland BENOIT et al.

**1/Reviewer 2**

**Comments:** The paper presents a set of experiments conducted in Jet Stirred Reactor (JSR) studying cool flame combustion of two atmospherically-relevant monoterpenes, a-pinene and limonene. The authors present the observed oxidation products in a graphical form. In addition, these products are compared to literature, where the oxidation of these monoterpenes were studied at atmospherically relevant conditions. The authors discuss the similarities and the percentage of the products that were formed in autoxidation reactions. I think that the results could be interesting to the atmospheric chemistry community. Below, I provide few general comments and many specific comments which the authors should address before the manuscript can be accepted.

**Answer:**

We thank the reviewer for his comments and suggestions which we tried to address in the revision

**2/Reviewer 2**

**Comment:** There are few places in the text where authors discuss the oxidation products that are formed in autoxidation. E.g., line 298: "Here, we propose a new approach which consists in assessing a set of molecules mainly resulting from autoxidation". In addition, in lines 269-276, it is said that the observed autoxidation products are "linked to each other by a single difference of 2 oxygen atoms", similar is in line 340. While I do not oppose the idea that these products are formed via some steps of autoxidation, the authors need to explain what they mean and what exactly their method is. From Figure 2b, I can see that all kinds of products are detected, not only the ones separated by two oxygens (i.e. sequential O2 additions). This is expected as the oxygen parity is lost if other reactions than autoxidation happen (as authors also discuss), but it does not explain how autoxidation was separated in this work. The arrows in figure 2b look like they were drawn arbitrarily. In addition, some explanation is needed in the main text how the percentage of autoxidation products was calculated, e.g. 73% for compounds detected by APCI.

**Answer:**

Some clarifications have been added concerning the molecules formed by autoxidation and the screening method.
We propose this modification line 73: Autoxidation will preferentially form chemical functions such as carbonyls, hydroperoxyl, or peroxyl. This large diversity of chemical functions will promote the formation of isomers. Nevertheless, the common point to these chemical compounds is the sequential addition of O2. Therefore, in a database, potential candidate products of autoxidation are easily identified by this sequential addition. The relevance of the tool was evaluated under different experimental conditions.

We also propose this modification line 269-272: In the OSc versus carbon number plot (Fig 2a), the vertical lines (at constant carbon number) are a first criterion for finding potential candidate products of

autoxidation. Figure 2b shows, for a fixed number of carbon and hydrogen atoms and a difference of two oxygen atoms, the potential candidate products of autoxidation connected by arrows whose color characterizes the oxygen parity. We can measure the extent of autoxidation for each carbon backbone in the OSc vs. O/C space. Under these criteria, we found that 73% of all chemical formulae are linked by a single difference of two oxygen atoms (which reflects an autoxidation mechanism).

**3/Reviewer 2**

**Comment: Introduction section**. It seems that the authors' previous paper (Benoit et al. 2021) is focused on comparing limonene oxidation products in JSR and at ambient conditions. Authors should specify in text if the dataset from current study is new or the same as in Benoit et al. 2021 and briefly explain how the current paper's comparison for limonene is similar or different.

**Answer**:

A modification has been made line 118: Compared to previous works, we have added the study of α-pinene oxidation to that of limonene and investigated the impact of ionization modes on the number of molecules detected and their chemical nature (unsaturation, oxidation rate). The size of the experimental and bibliographic databases has been increased by more than 50%, in particular by adding data specific to autoxidation (Krechmer et al. (2016); Tomaz et al. (2021) and references on α-pinene)

**4/Reviewer 2**

**Comment**: The section on HPLC results at the end of the manuscript is confusing. Can the authors please explain more clearly in the text how it is relevant and what are the main findings complementing the results of current manuscript, even if the results are negative (i.e. coelution was observed

**Answer**:

See answer 31

**5/Reviewer 2**

**Comment**: Figure 3 presents the conceptual schematic of autoxidation in combustion and atmosphere for oxidation with OH. This figure is only once mentioned in the text (line 276) to show the change of oxygen parity. While a simple figure can help to represent this, I think there are problems with the schemes represented. First of all, only OH pathway is considered, while in atmospheric oxidation data, this paper looks at many ozonolysis experiments. Secondly, H-abstraction pathway is shown also as an atmospheric pathway, while OH addition is prevalent for oxidation of monoterpenes (e.g. Berndt et al. 2016). This clearly changes the oxygen parity (odd vs even), which is again different if ozonolysis products are considered. It would be great if the authors specified how the pathways presented in this schematic are used in the paper and perhaps clarify if these are just subset of reactions that are possible and why authors chose these. Also, please include some references in the schematic.

**Answer**:

We propose this modification line 276: Figure 3 illustrates one of the reaction mechanisms (OH pathway) of oxygen parity change in the case of an autoxidation reaction. It should be noted that other reaction mechanisms can change oxygen parity. In addition, the OH pathway in ozonolysis is not the predominant one. (Berndt et al 2016; 10.1021/acsearthspacechem.9b00035).

Bibliographic references are added to the caption of revised Figure 3

**6/Reviewer 2**

**Comment**: The authors should include a list of observed products in JSR experiments to Supplementary.

**Answer**:

A database with all chemical species identified in combustion (JSR) is added to the Supplementary.

**Abstract**

**7/Reviewer 2**

**Comment**: Line 24-25: "We built an experimental database" – where this database will be deposited? Will it include list of all the compounds?

**Answer**:

See previous answer

**8/Reviewer 2**

**Comment**: Line 28: "a subset of chemical formulas is common to all experiments independently of experimental conditions." – I don't think this is true for a-pinene case where surprisingly there were no compounds in common, Figure 5a?

**Answer**:

This result confirms the need to use both ionization sources. It is noted in the literature cited in this study that analyses performed with an APCI source do not show data common to all previous experiments listed in Table 2. The addition of our dataset cannot change this result. The similarities appear only in case-by-case comparison. With the HESI source, similarities appear in the literature data (Fig. 4b) and are consolidated by our experiences. This difference could be explained by the fact that the detection of higher molecular weight chemicals is easier using HESI (Parr et al., Journal of Chromatography B, 10.1016/j.jchromb.2018.05.017) (Fig. 6).

We propose these modifications line 28: Data analysis (in HESI mode) indicated that a subset of chemical formulae is common to all experiments independently of experimental conditions.

**9/Reviewer 2**

**Comment**: Line 59: "highly oxidized molecules" - HOMs are defined as "highly oxygenated organic molecules" as per references provided in that line.

**Answer**: done

**Comment**: Line 63: I would absolutely list Iyer et al. 2021 as a reference for "few hundredths of a second", I don't think other studies studied such short time scales. I suggest authors rethink the order of references in this paragraph to include a) original references first and b) to put the relevant references in the right sentences.;

**Answer**: Indeed, the studies cited give a time scale of less than a second, without further precision. The reference Iyer et al. 2021 is cited in this revision

**10/Reviewer 2**

**Comment**: Line 77: "to name a few" – please remove this. Which other radicals?

**Answer**:

We propose this modification line 77: In laboratory studies conducted under simulated atmospheric conditions, oxidation occurs at near-ambient temperatures (250-300 K), at atmospheric pressure, in the presence of ozone and/or •OH radicals (Table 2), used to initiate oxidation with low initial terpene concentrations.

**11/Reviewer 2**

**Comment**: Line 79: "so as to compensate for the absence of ozone". Can authors clarify what they mean?

**Answer**:

We propose this modification line 79: In addition to the increase in temperature, the initial concentrations of the reagents are generally higher compared to the atmospheric conditions, in order to initiate the oxidation with $O_2$, which is much slower than that involving ozone.

**12/Reviewer 2**

**Comment**: Line 86: For Huang et al. 2018 reference, you have to mention that it was for particle phase composition, because in the text so far refers to the gas-phase. Further sentences are about aerosol particles, but a smoother connection is required here.

**Answer**:

We propose this modification line 86: Similarly, the experiments of Huang et al. performed at different temperatures (223 K and 296 K) and precursor concentrations (α-pinene 0.714 and 2.2 ppm) suggested that the physicochemical properties, such as the composition of the oligomers (at the nanometer scale), can be affected by a variation of temperature

**13/Reviewer 2**

**Comment**: Line 89: "oligomerization, addition, or accretion" I do not know what is the difference between these and how they relate to earlier phrase. Do you mean nucleation or condensation? Because while there can be accretion products, they are not yet aerosol particle. Please clarify.

**Answer**:

The words oligomerization, functionalization, or accretion are the different terms used in the literature to describe the formation and growth of aerosols. The words addition has been removed. Two references have been added to support this sentence. M. Camredon et al. Atmos. Chem. Phys., 10, 2893–2917, 2010, 10.5194/acp-10-2893-2010; Meusinger et al. Atmos. Chem. Phys., 17, 6373–6391, 2017, 10.5194/acp-17-6373-2017.

**14/Reviewer 2**

**Comment**: Lines 103-105: slightly unclear sentence, at least for the first read. Also, what does CxHyO1+15 mean?

**Answer**:

We have developed a software tool that allows, from the mass spectra, to edit immediately all the graphic tools.

Also, what does CxHyO1+15 mean? This is a typo: it should read "$C_xH_yO_{1\text{ to }15}$"

**15/Reviewer 2**

**Comment**: Line 187: I don't think there is "-O" in DBE formula. Please correct.

**Answer**:

Corrected.

**16/Reviewer 2**

**Comment**: Line 224-225: "appropriate for studying products with a large number of unsaturation (DBE > 5), probably related to the increase in the number of hydroxyl groups". Can authors clarify how the addition of OH groups changes DBE, because I do not think there is a relationship.

**Answer**:

We propose this modification line 224-225: probably due to the coexistence of carbonyl and hydroperoxide functional groups in autoxidation products.

**17/Reviewer 2**

**Comment**: Line 253: "with maximum counts recorded for 10 O-atoms (Fig. 1c)" – Did you mean most of chemical formulae (y axis of Fig 1c) and not maximum counts (as in signal strength)?

Based on the figure, I can tell that it is with 9 O atoms, not 10. Also, x-axis of both Fig 1b and c are somehow shifted, so it is hard to see.

**Answer**:

We propose this modification line 253: In the case of limonene, with a HESI source, chemicals with an oxygen number of up to 15 were detected. Most of the chemical formulae recorded had 8-10 O-atoms (Fig. 1c), whereas for α-pinene the products with >8 O-atoms were much less (Fig. 1b)

Shift of the axes has been corrected

**18/Reviewer 2**

**Comment**: Figure 2a: What is the hypothesis for forming compounds with more than 20 carbon numbers? To me, it seems that there are many oxidation steps that take place after which also accretion products form RO2 radicals and form another accretion products. Are these compounds typically observed in combustion? This would be good to clarify also where the datasets compared to atmospherically relevant oxidation studies.

**Answer**:

This study shows that both in our combustion experiments and in atmospheric chemistry, it is possible to observe oxidized chemical compounds (CxHyOz) with 20 carbon atoms. We present this hypothesis and the work of Wang et al, 2021, 10.1038/s42004-020-00445-3 confirms such observation, and describes the accretion phenomenon, even if the presence of these chemical compounds in combustion, seems to be lower. We present these chemical compounds from both atmospheric and combustion datasets with a skeleton of 20 carbon atoms in Figure 6 ).

**19/Reviewer 2**

**Comment**: Figure 2b: This figure reveals that the observed oxidation products have composition of C10H4O9 and so on (with very little H), how is this possible?

**Answer**:

We do not have any specific information on this type of compound. We chose to include these chemical formulae because their detection is recurrent. However, they have not been discussed in the results.

**20/Reviewer 2**

**Comment**: You are testing accretion hypothesis with OSc plot, but isn't the fact the compounds have 11 carbons already shows you that it is an accretion product? Please clarify the text.

**Answer**:

We propose this modification line 261: Indeed, the presence of chemical compounds with 11 carbon atoms can be explained by an accretion phenomenon (Wang et al., 10.1038/s42004-020-00445-3), but the advantage of this OSc-nC space representation (Kroll et al., 10.1038/nchem.948) is to study this phenomenon on all the data.

**21/Reviewer 2**

**Comment**: Lines 266-269: Is there any advantage to add the classes of organic aerosols in text and on Figure 2a?

**Answer**:

The representation of the classes allows to better understand the evolution of the volatility of the chemical compounds formed. This tool is also used in the literature and thus provides a framework that is easier to read.

**22/Reviewer 2**

**Comment**: Line 293: Perhaps there is a better reference that can be used here?

**Answer**:

We suggest adding these three references: Crounse et al., 2013,10.1021/jz4019207; Jokinen et al., 2014, 10.1002/anie.201408566; Lyer et al., 2021, 10.1038/s41467-021-21172-w.

**23/Reviewer 2**

Lines 293-295. Please rephrase, it is unclear what authors mean. For instance, what does "other carbon skeletons are also concerned by autoxidation" mean?

**Answer**:

We propose this modification line 295-296: However, previous studies on the propagation of this reaction mechanism have mainly focused on the initial steps of autoxidation without screening all identified chemical formulations for potential autoxidation products. It is therefore useful to assess the proportion of possible autoxidation products among the total chemical species formed.

**24/Reviewer 2**

**Comment**: Line 304: It seems that the authors use only 4 of the limonene studies. Perhaps there is no reason to include all of them in the table, especially since the table appears to be the same as in Benoit et al. 2021.

**Answer**:

The number of bibliographic citations compared to Benoit et al. 2021 is higher. The number of chemical formulae and the presence of liquid chromatography characterization have been added. The collection of these data is time-consuming; we believe that this could be useful to readers to have these available and up to date in the paper.

**25/Reviewer 2**

**Comment**: Line 308: "files provided" – no files or data is actually shown in SI. Please include the list of compounds.

**Answer**:

The database will be added to the final version of the Supplementary.

**26/Reviewer 2**

**Comment**: Lines 311-312: I think this is incorrect to assume that the method of analysis does not affect which species are detected. In fact, it is known that NO3- chemical ionization is selective to highly oxygenated compounds and will not detect some species (e.g. Ehn et al. 2014, Riva et al. 2019). As a result, studies done with NO3- ionization have almost no to little overlap in Fig. 4.

**Answer**:

That's right. The sentence was probably confusing. We propose this modification, line 311-312: We considered that the ionisation mode used in mass spectrometry did not modify the nature of the chemical species but only the relative detection of ions, because of the type of ionization used and the sensitivity of the instruments

**27/Reviewer 2**

**Comment**: Figure 4a: How is it possible that not a single species is common between these studies?

**Answer**:

There are species common to our study and the studies cited. As you mentioned, one hypothesis may be the selectivity of the NO3 ionisation mode. Please see Answer to comment #8

**28/Reviewer 2**

**Comment**: Lines 328-330: This sentence is a bit complicated. Please rephrase.

**Answer**:

We propose this modification line 328-330: For the oxidation of limonene, the four selected studies identified 1434 chemical formulae. Among these studies, the experiments by Walser et al. were performed with both (+) and (–) ionisation modes. In contrast to the α-pinene case, the selected studies for limonene were performed with similar ionization sources, which probably contributed to increased data similarity.

**29/Reviewer 2**

**Comment**: Lines 342-345: Can authors explain how "relationship to autoxidation" determined and these results obtained?

**Answer**:

These 78 chemical formulae common to the study set have no reason to be linked to each other, except that they result from the oxidation of limonene, but with different experimental conditions. Nevertheless, it is assumed that autoxidation could be a common reaction pathway by verifying a difference of two oxygens between them.

The percentage characterises the share of chemical formulae with a single difference of two oxygens and included in this set. In the case of our hypothesis, autoxidation would therefore be one of the dominant oxidation mechanisms.

**30/Reviewer 2**

**Comment**: Lines 357-358. Presenting here the results in percent is misleading. Maybe just report number of chemical formulas that are common, as done in previous paragraph.

**Answer**:

In addition to percentages we report number of chemical formulae that are common: (Deng et al., 2021) 69% (243), (Meusinger et al., 2017) 46%, (71), …

**31/Reviewer 2**

**Comment**: Section 4.3.2: "Detailed analysis". I am not sure which questions the authors try to answer with this section. Also, main compounds in atmospheric chemistry are listed here. Why? Can isomers be marked in Fig. 8? More description is needed if this section is relevant.

**Answer**:

The paragraph has been completely rewritten with more details on the objectives. We propose this modification line 398:

4.3.2 Identification of common isomers.

We identified a set of chemical formulae common to both atmospheric and combustion chemistries and suggested that this might result from an autoxidation mechanism. We identified several chemical formulae within this dataset that differ by two oxygen atoms on the same skeleton. Focusing on the early stages of limonene oxidation, there are several sequential two-oxygen additions to the chemical formulae $C_{10}H_{16}O_2$ and $C_{10}H_{16}O_3$ with the two oxygen parities described in Figure 3. Tables 3 present the identified chemical formulae with information on the Venn index. The index for combustion is the rightmost (xxxx1).

Tableau 3. Sequential additions of two oxygens to the chemical formulae $C_{10}H_{16}O_2$ and $C_{10}H_{16}O_3$ present in the common set of 78 chemical formulae

| First stages of oxidation | 1st addition | 2nd addition | 3rd addition | 4th addition | 5th addition |
|---|---|---|---|---|---|
| $C_{10}H_{16}O_2$ *(10101)* | $C_{10}H_{16}O_4$ *(11101)* | $C_{10}H_{16}O_6$ *(11111)* | $C_{10}H_{16}O_8$ *(01011)* | $C_{10}H_{16}O_{10}$ *(00011)* | $C_{10}H_{16}O_{12}$ *(00010)* |
| $C_{10}H_{16}O_3$ *(11101)* | $C_{10}H_{16}O_5$ *(11111)* | $C_{10}H_{16}O_7$ *(11111)* | $C_{10}H_{16}O_9$ *(01011)* | $C_{10}H_{16}O_{11}$ *(00010)* | |

This result of sequential additions of two oxygens is also observed for other chemical formulae of this common dataset. Therefore, it questions the possibility that these two atmospheric and combustion chemistries develop autoxidation mechanisms with common isomers.

In order to verify this possibility, considering the differences between limonene and α-pinene, we analyzed by UHPLC-HRMS the chemical compounds $C_{10}H_{16}O_2$ for limonene and $C_{10}H_{16}O_3$ for α-pinene in the samples from the combustion experiments. For limonene and α-pinene, considering the availability of standards from suppliers, we selected limonoaldehyde and pinonic acid, respectively. These two isomers of $C_{10}H_{16}O_2$ and $C_{10}H_{16}O_3$ are among the most frequently reported products in atmospheric chemistry studies (Table 4). Our study shows same retention times for these standards and isomers detected in combustion samples (Fig S4). This result is more evident for limonoaldehyde (11.5 min) than for acid pinonic (3.9 min). In addition, we detected the presence of -OH or -OOH groups by H/D exchange with D2O for these two chemical formulae. Unfortunately, coelution did not fully allow exploiting MS/MS fragmentation carried out on the two chemical formulae, and to formally identify the two compounds. There is still a lot of characterization work to be done, but the hypothesis of common isomeric products formed through an autoxidation mechanism operating in atmospheric and low-temperature combustion conditions seems to be confirmed.

**Table 4.** Isomers of α-pinene and limonene oxidation reported in the literature.

| | $C_{10}H_{16}O_2$ | | $C_{10}H_{16}O_3$ | |
|---|---|---|---|---|
| α-pinene | Pinonaldehyde | (Fang et al., 2017) | Pinonic acid | (Fang et al., 2017;Ng et al., 2011;Meusinger et al., 2017) |
| | Hydroxyketone | (Fang et al., 2017) | hydroxy pinonaldehydes | (Fang et al., 2017;Meusinger et al., 2017) |
| Limonene | Limononaldehyde | (Fang et al., 2017;Walser et al., 2008;Bateman et al., 2009) | Limononic acid | (Fang et al., 2017;Witkowski and Gierczak, 2017;Hammes et al., 2019;Walser et al., 2008;Bateman et al., 2009;Warscheid and Hoffmann, 2001) |
| | 4-isopropenyl-methylhydroxy-2-oxocyclohexane | (Fang et al., 2017) | 7-hydroxy-limononaldehyde | (Fang et al., 2017;Walser et al., 2008;Bateman et al., 2009;Meusinger et al., 2017) |

**33/Reviewer 2**

**Comment**: Line 441- what is "COVs"?

**Answer**:

Replaced by VOCs.

**34/Reviewer 2**

**Comment**: Line 452: Is this provided data from Tomaz et al. 2021? If yes, please indicate so and remove the word "his".

**Answer**:

Yes, the reference was corrected.

**Supplementary**

**35/Reviewer 2**

**Comment**: I think 3D figures contain too much information which is hard to interpret. I would suggest to consider 2D plots that can show relevant information, for instance, DBE vs OSc or similar.

**Answer**:

We have tested different solutions for the graphical representation of the data. Unfortunately, in the 2D views, due to the amount of data (almost 3000 points) some information is not visible. In the Osc vs DBE solution, this is the case. The offset of the overlaid points also remains unsatisfactory (see the two graphs below). We propose to add the first one in Supplements. The graph bellow is included in the revised Supporting

We propose this modification line 374: and an OSc vs DBE graph

Without offset

[Figure]

With offset

[Figure]

**Technical and minor comments**

**36/Reviewer 2**

**Comment**: Lines 52-55: H-Shift formulas- can you format them as a column for easier read? (like equations)

**Answer**:

Modified accordingly:

$R^\bullet + O_2 \rightleftarrows ROO^\bullet$ (first $O_2$-addition)
$ROO^\bullet \rightleftarrows {}^\bullet QOOH$ (H-shift)
${}^\bullet QOOH + O_2 \rightleftarrows {}^\bullet OOQOOH$ (second $O_2$-addition)
${}^\bullet OOQOOH \rightleftarrows HOOQ^{\bullet\prime}OOH$ (H-shift)
$HOOQ^{\bullet\prime}OOH + O_2 \rightleftarrows (HOO)_2Q'OO^\bullet$ (third $O_2$-addition)
$(HOO)_2Q'OO^\bullet \rightleftarrows (HOO)_2Q'''OOH$ (H-shift)
$(HOO)_2Q'''OOH + O_2 \rightleftarrows (HOO)_3Q'' OO^\bullet$ (fourth $O_2$-addition), etc.

**37/Reviewer 2**

**Comment**: Line 56: Is abbreviation "HOPs" needed? It is only used in Abstract and here and is not a common abbreviation generally used in the literature, to my knowledge.

**Answer**:

Abbreviation removed

**38/Reviewer 2**

**Comment**: Line 55: "There, the formation of HOPs was mainly attributed to autoxidation reactions." – I don't understand "where"

**Answer**:

In cold flame combustion, autoxidation mechanisms are enhanced. The literature cited in this paragraph and our work confirm this result.

**39/Reviewer 2**

**Comment**: Line 58&60: Please add Ehn 2014 (Nature) to the references for HOM and autoxidation.

**Answer**:

Corrected.

**40/Reviewer 2**

**Comment**: Line 161: "I was"-> It was

**Answer**:

Corrected.

**41/Reviewer 2**

**Comment**: Line 164: "these two isomers" – do you mean here two monoterpene isomers or something else?

**Answer**:

Corrected: two monoterpene isomers

**42/Reviewer 2**

**Comment**: Line 246: "limited to 10 oxygen atoms" – do you mean "limited to compounds with 10 oxygen atoms or lower"?

**Answer**:

Corrected: limited to compounds with 10 oxygen atoms or lower

**43/Reviewer 2**

**Comment**: Line 307: A year is missing in the reference.

**Answer**:

Corrected: 2001

**44/Reviewer 2**

**Comment**: Table 2. In column titles "ionization/source", there are inconsistencies. For instance, Jokinen et al. 2015 used NO3- ionization, but it is not mentioned. Same goes for Hammes et al. 2019. Please check that information in this table is correct and consistent. Please also check online/offline metric. I am not familiar with offline PTR-TOF method, for instance.

**Answer**:

Corrected: Jokinen et al. 2015 used NO3- ionization,

Hammes et al. 2019, used a 210Po alpha emitter to produce acetate reagent ions.

**Supplementary**

**45/Reviewer 2**

**Comment**: Table S1: Please check the legend is correct in the figures located at the last row. It seems that "a-pinene only" and "limonene-only" are inconsistent between two graphs. In Table S2 they seem to be correct.

**Answer**:

The legend to Figure S1 has been corrected

**References**

References have been corrected

**Dear Reviewer (3),**

Thank you for your comments and suggestions. We have taken them into account in the revised manuscript.

In this document, you will find changes made in response to your comments

Best regards,

Roland BENOIT et al.

**1/Reviewer 3**

**Comments:** This is a review of "On the formation of highly oxidized pollutants by autoxidation of terpenes under low temperature combustion conditions: the case of limonene and α- pinene" by Benoit et al., submitted to ACP. This is an interesting paper that blends the atmospheric chemistry and combustion communities. Specifically, this work has implications for low-temperature combustion of monoterpenes, as would occur during smoldering wildfire conditions.

**Answer**:

We thank the reviewer for his comments and suggestions which we tried to address in the revision

**2/Reviewer 3**

**Comments:** That being said, it would've been nice if the authors could more strongly detail that link. For example, in Table 2 the comparisons are all with papers that look at ozonolysis of monoterpenes, and there is not much comparison or discussion on what is seen during wildfire sampling

**Answer**:

One of the difficulties of this study was to select usable databases. We studied and cited the literature on forest fires or biomass burning (Smith et al 2009; Hatch et al. 2019; Gilman et al. 2015). A new bibliographic reference was added: Schneider et al., ACS Earth Space Chem. 2022, 10.1021/acsearthspacechem.2c00017.

There is not much comparison or discussion of what is seen during forest fire sampling because field experiments are not appropriate for the present investigation as a strict distinction on the origins of the chemical compounds observed cannot be assessed.

We propose this modification line 47: But, field experiments are not appropriate for the present investigation because a strict distinction on the origins of the chemical compounds observed cannot be assessed. For example, literature works and reviews (Hu et al., Int Journal of Wildland fire, 2018, 27, 10.1071/WF17084; Popovicheva et al., Aerosol and Atmos Chemistry, 2019, 19, 10.4209/aaqr.2018.08.0302; Prichard et al.; Int Journal of Wildland fire, 2020, 10.1071/WF19066) present field measurements from smoldering fires which couldn't be used here. Our combustion database will be added to the Supplements to allow the scientific community which works on this topic to exploit our results.

**Specific Comments:**

**3/Reviewer 3**

**Comments:** Line 43: reference needed for fires getting more frequent and intense

**Answer**:

Reference added: Burke et al., PNAS, 2021, The changing risk and burden of wildfire in the United States, 10.1073/pnas.2011048118

**4/Reviewer 3**

**Comments:** Line 146: when were samples analyzed? I.e., what was the shortest time they were in the freezer, if it all? Couldn't there be prompt degradation, followed by no further degradation from the shortest analysis time up to the one month maximum?

**Answer**:

The shortest time is a few minutes, without freezer storage. For the samples stored in the freezer, we observed a decrease in the intensity of recorded MS signal over time, mainly after thirty days. This decrease varies from 10% to 30%. Nevertheless, we could not observe significant impact on these results, since no quantification could be carried out.

**5/Reviewer 3**

**Comments:** Line 161: "I" should be "It"?

**Answer**: corrected

**6/Reviewer 3**

**Comments:** Line 162: was this utilized as the background? Could be nice to add an SI figure of this mass spectrum for proof.

**Answer**: The spectrum of the reference limonene was not used as background. To address this point, a comparison of the mass spectra of reference and oxidized limonene is added to the Supplements.

[Figure]

**7/Reviewer 3**

**Comments:** Line 175: "combustion-specific" isn't appropriate here, since your argument is that combustion generates similar products as oxidation. Maybe "combustion-relevant"?

**Answer**:

"Combustion-relevant' is indeed more appropriate. This is corrected

**8/Reviewer 3**

**Comments:** Line 189: how many/ what percentage of your identified peaks had odd numbers of hydrogens? I would think this would be an insignificant amount?

**Answer:**

There are very few, a few dozen out of several thousand peaks detected. We attribute these chemical formulae with an odd number of hydrogens to radicals that should not be detected with our Orbitrap.

**9/Reviewer 3**

**Comments:** Line 197: this sentence could be re-written to be clearer. "low temperature should be optimal"—do you mean efficient? Because you're not talking about quantity of -OOH, but rather yields? I think also "even if the conversion…" should be "even though the conversion", because you already stated only 1% is oxidized (on line 139)?

**Answer:**

The sentence is corrected - line 197: Under these conditions, the formation of peroxides by autoxidation at low temperature should be efficient (Belhadj et al., 2021c), even though the conversion of the fuels remains moderate.

**10/Reviewer 3**

**Comments:** Line 203: how are you getting relative concentrations for every molecule? Shouldn't this be reported in relative signal, unless you've calibrated for every peak?

**Answer:**

Indeed, only the intensity of the peaks (relative intensity) is considered and there are no relative concentrations due to lack of standards for MS calibration.

**11/Reviewer 3**

**Comments:** Line 204: does "highest mass peak" mean the molecule at the highest mass ie molecular weight, or the molecule with the highest measured mass ie concentration?

**Answer:**

This has already been indicated on line 167. We removed the sentence and edit Line 167 by replacing "highest mass peak" by "highest MS signal"

**12/Reviewer 3**

**Comments:** Line 215-216: I don't think this wording is strictly correct, and would reword to something more like "i.e. using both polarities captures between 30-50% more molecular species".

And the next sentence to something closer to "utilizing both polarities is helpful in identifying a larger quantity of species". It's nearly certain that even utilizing two instruments and two polarities you won't be capturing all the molecular species. For example, Isaacman-VanWertz et al. (2018, Nature Chemistry, https://www.nature.com/articles/s41557-018-0002-2) were not able to achieve carbon closure for the entirety of their experimental time, even with numerous instruments.

**Answer**:

The sentence is corrected - line 215: For a given ionization source, ~ 50% of the chemical formulae are common to both polarities, i.e., using both polarities captures between 30-50% more molecular species (since some of them are ionized under a single mode (+ or –) depending on their chemical structure). Utilizing both polarities is helpful in identifying a larger quantity of species. The HESI source data were compared to the APCI data (Supplement, Tables S1 and S2), showing an increased number of chemical formulae detected by 20 to 30%.

**13/Reviewer 3**

**Comments:** Line 226: I assume you're implying because the hydroxyl groups are sticky/make the molecule lower volatility. I suggest stating this explicitly.

**Answer**:

The sentence is corrected - line 224-225 In addition to these observations, we noted that HESI is more appropriate for studying products with a large number of unsaturation (DBE > 5), probably related to the increase in the number of hydroperoxide and carboxyl groups along with heated source favor vaporization of low volatility compounds.

**14/Reviewer 3**

**Comments:** Line 229: should "tight" be "tied" and maybe "functions" should be "functionality" or "functional groups"?

**Answer**:

The sentence was corrected - line 229: The most appropriate ionization polarity to be used is tied to functional groups present in products to be detected

**15/Reviewer 3**

**Comments:** Line 243: I don't think this definitively shows that the terpenes are ring opened, and would use less conclusive language, like "formulas obtained suggest that the oxidation of.."

**Answer**:

Indeed the word "suggest" is more appropriate: corrected

**16/Reviewer 3**

**Comments:** Line 246-249: do you expect this to be the same, or has this previously been observed, for b-pinene which also has an exocyclic double bond? Could it instead be because limonene has two double bonds, versus only 1 in a-pinene?

**Answer**:

In the literature, the reactivity of limonene is discussed in terms of exo- versus endo-double bond and in terms of multiple double bond in the case of limonene versus α-pinene which has a single double bond (Pospisilova et al, Sci. Adv. 2020, 10.1126/sciadv.aax8922; Kundu et al. Chem. Phys, 2012, 10.5194/acp-12-5523-2012). The presence of two double bonds likely favors the formation of heavy oxygenated products.

**17/Reviewer 3**

**Comments:** Line 255: it looks like the distribution is centered on 9 oxygen's, not 11?

**Answer**:

The error was corrected

**18/Reviewer 3**

**Comments:** Figure 2A: the hashed colored circles for molecule classification are difficult to differentiate. Suggest using 4 colors that are more different.

**Answer**:

The figure 2A has been modified for more visibility.

[Figure]

**19/Reviewer 3**

**Comments:** Figure 2B: The meaning of the arrows is confusing, is autoxidation only indicated across the few dots that each arrow touches? Are the even number and odd number of oxygen arrows supposed to be indicating the different slopes of behavior or just the individual dots they touch? It's also not clear why there are multiple colors for the autoxidation arrows?

**Answer**:

Autoxidation is represented for a few points. This is an illustration of the possible chemical reaction pathways for two oxygen parities. The curved arrows indicate only the individual points they touch. Figure 2b has been modified with only two colours. The two initial arrows characterise the early stages of oxidation from limonene.

[Figure]

**20/Reviewer 3**

**Comments:** Line 304: can you quantify "enough chemical products"? It's not clear how and why you picked these 15 studies out of the truly immense number of a-pinene oxidation studies. Why were only ozonolysis studies (line 310) chosen?

**Answer**:

The literature on this subject is very extensive indeed. Our choice was driven by the choice of databases and the potential existence of autoxidation mechanisms. The ozonolysis of terpenes and more particularly of alpha pine and limonene are among the most documented studies with recent descriptions of autoxidation mechanisms, associated with databases.

**21/Reviewer 3**

**Comments:** Line 329: a reference for limonene having more accretion products would be helpful, or just a reference to Figure 1.

**Answer**:

A bibliographic reference was added line 329: Jokinen et al. Angew. Chem. Int. Ed. 2014, 10.1002/anie.201408566

**22/Reviewer 3**

**Comments:**

Figures 4 & 5: the purpose of the binary numbers is not clear. These numbers also come up in section 4.3.2. There's a short sentence in the figure 4 caption, but more elaboration in the main text is probably needed.

**Answer**:

Clarifications have been made to the nomenclature of the Venn diagrams in the legend (Fig. a): Each digit indicates a study, the value of the digit characterises the presence (value 1) or absence (0) of the study in the result. e.g.: 23 chemical formulae (0101) (fig. 4a) are common to the studies of Deng et al; (0001) and Meusinger et al. (0100)

**23/Reviewer 3**

**Comments:** Lines 399 & 406: "analyzes" should be "analysis"

**Answer**: corrected

**24/Reviewer 3**

**Comments:** Section 4.3.2: this section on HPLC doesn't feel fully fleshed out. It's description in the methods section also feels like an after-thought. This may want to be moved to the SI, or expanded upon.

**Answer**:

The paragraph has been completely rewritten with more details on the objectives. We propose this modification line 398:

4.3.2 Identification of common isomers.

We identified a set of chemical formulae common to both atmospheric and combustion chemistries and suggested that this might result from an autoxidation mechanism. We identified several chemical formulae within this dataset that differ by two oxygen atoms on the same skeleton. Focusing on the early stages of limonene oxidation, there are several sequential two-oxygen additions to the chemical formulae $C_{10}H_{16}O_2$ and $C_{10}H_{16}O_3$ with the two oxygen parities described in Figure 3. Tables 3 present the identified chemical formulae with information on the Venn index. The index for combustion is the rightmost (xxxx1).

Tableau 3. Sequential additions of two oxygens to the chemical formulae $C_{10}H_{16}O_2$ and $C_{10}H_{16}O_3$ present in the common set of 78 chemical formulae

| First stages of oxidation | 1st addition | 2nd addition | 3rd addition | 4th addition | 5th addition |
|---|---|---|---|---|---|
| $C_{10}H_{16}O_2$ *(10101)* | $C_{10}H_{16}O_4$ *(11101)* | $C_{10}H_{16}O_6$ *(11111)* | $C_{10}H_{16}O_8$ *(01011)* | $C_{10}H_{16}O_{10}$ *(00011)* | $C_{10}H_{16}O_{12}$ *(00010)* |
| $C_{10}H_{16}O_3$ *(11101)* | $C_{10}H_{16}O_5$ *(11111)* | $C_{10}H_{16}O_7$ *(11111)* | $C_{10}H_{16}O_9$ *(01011)* | $C_{10}H_{16}O_{11}$ *(00010)* | |

This result of sequential additions of two oxygens is also observed for other chemical formulae of this common dataset. Therefore, it questions the possibility that these two atmospheric and combustion chemistries develop autoxidation mechanisms with common isomers.

In order to verify this possibility, considering the differences between limonene and α-pinene, we analyzed by UHPLC-HRMS the chemical compounds $C_{10}H_{16}O_2$ for limonene and $C_{10}H_{16}O_3$ for α-pinene in the samples from the combustion experiments. For limonene and α-pinene, considering the availability of standards from suppliers, we selected limonoaldehyde and pinonic acid, respectively. These two isomers of $C_{10}H_{16}O_2$ and $C_{10}H_{16}O_3$ are among the most frequently reported products in atmospheric chemistry studies (Table 4). Our study shows same retention times for these standards and isomers detected in combustion samples (Fig S4). This result is more evident for limonoaldehyde (11.5 min) than for acid pinonic (3.9 min). In addition, we detected the presence of -OH or -OOH groups by H/D exchange with D2O for these two chemical formulae. Unfortunately, coelution did not fully allow exploiting MS/MS fragmentation carried out on the two chemical formulae, and to formally identify the two compounds. There is still a lot of characterization work to be done, but the hypothesis of common isomeric products formed through an autoxidation mechanism operating in atmospheric and low-temperature combustion conditions seems to be confirmed.

**Table 4.** Isomers of α-pinene and limonene oxidation reported in the literature.

| | $C_{10}H_{16}O_2$ | | $C_{10}H_{16}O_3$ | |
|---|---|---|---|---|
| α-pinene | Pinonaldehyde | (Fang et al., 2017) | pinonic acid | (Fang et al., 2017;Ng et al., 2011;Meusinger et al., 2017) |
| | hydroxyketone | (Fang et al., 2017) | hydroxy pinonaldehydes | (Fang et al., 2017;Meusinger et al., 2017) |
| Limonene | limononaldehyde | (Fang et al., 2017;Walser et al., 2008;Bateman et al., 2009) | limononic acid | (Fang et al., 2017;Witkowski and Gierczak, 2017;Hammes et al., 2019;Walser et al., 2008;Bateman et al., 2009;Warscheid and Hoffmann, 2001) |
| | 4-isopropenyl-methylhydroxy-2-oxocyclohexane | (Fang et al., 2017) | 7-hydroxy-limononaldehyde | (Fang et al., 2017;Walser et al., 2008;Bateman et al., 2009;Meusinger et al., 2017) |

---

## Author Response (AR2)

Dear Editor,

Thank you for your comments and suggestions. We have taken them into account in the revised manuscript.

In this document, you will find changes made in response to reviewer 2 comments

Best regards,

Roland BENOIT et al.

**Reviewer 2**

**Comments:**

The authors clarified few points raised by the reviewers. However, there are still some shortcomings on discussing the autoxidation and composition of observed compounds with HESI/APCI. There are still too many unclear points to justify acceptance for publication, some of which were brought up also by other reviewers.

I am still puzzled how authors came to the conclusions regarding autoxidation. It is unclear to me how the autoxidation products were selected and how other products were confirmed not to form in autoxidation. This makes also the arrows on Figure 2b confusing. Indeed, some studies before relied on the O2 difference, but it was also supported by the relative abundance of the compounds (for instance, only even-oxygen compound were present) and quite often was only applied to products with larger oxygen number.

**Answer**:

First, one should remember autoxidation was first described in 'combustion studies' performed at relatively low-temperature (<800K); see Introduction for references going back to 1952. What was not considered in most 'combustion' studies was the formation on highly oxidized products which have been described in publications relative to 'atmospheric' oxidation studies and named HOMs. In 2017, one of the authors of the present work published in PNAS a study demonstrating the formation of highly oxidized products through autoxidation (https://doi.org/10.1073/pnas.1707564114). Several other studies by ZD Wang et al. and Belhadj et al. followed (some references cited in the present paper). Thus, there is no clear reasons to be 'puzzled when reading here we could form and detect autoxidation

products. Our strength, compared to what was done with TOF MS (resolution ~4,000), is the use of a much better sensitive and precise MS instrument (Orbitrap with resolution of 120,000).

The products were selected based on the autoxidation scheme (Fig. S3) given in the supplementary. There, one can see the formation routes to $C_{10}H_{14}O_{2-10}$ and $C_{10}H_{14}O_{3-11}$. These molecular formulas were detected in our experiments as show in fig. 2b. The others formulas presented in fig. 2b should result from others oxidation pathways. Indeed, products with chemical formulas with H≥16 cannot derived from autoxidation pathways described in Fig S3. However, other pathways (Fig. S5) can yield such species, e.g., through the initial addition of OH on terpenes double bonds followed by O2 addition and autoxidation of products.

Fig. 2b as been revised: arrows connecting molecular formulas have been removed since these species are not directly connected (Fig. S3) whereas the radicals generated these stable species are connected by an autoxidation pathway (e.g., $C_{10}H_{15}O_3 \rightarrow C_{10}H_{15}O_5 \rightarrow C_{10}H_{15}O_7 \rightarrow C_{10}H_{15}O_9 \rightarrow \ldots$, $C_{10}H_{15}O_2 \rightarrow C_{10}H_{15}O_4 \rightarrow C_{10}H_{15}O_6 \rightarrow C_{10}H_{15}O_8 \rightarrow \ldots$)

[Figure]

Two new figures (Fig S3 and S5) have been added to describe the reaction pathways that can form the detected molecules.

[Figure]

**Fig. S3:** Molecular forms resulting from an autoxidation mechanism of limonene: a) even number of oxygens b) odd number of oxygens.

**Fig. S5:** Some proposed formation routes to C10H16Ox species via autoxidation

The chemical formulas detected (for both oxygen parities) and their relative intensity were specified in Table 2. Chemical formulas are highlighted in red in the Supplementary database.

| | Limonene | Alpha-pinene |
|---|---|---|
| **Even number of oxygen atom** | | |
| **hydroxyketone** | $C_{10}H_{14}O_2$ (1.61442E+7) | $C_{10}H_{14}O_2$ (4.54785E+7) |
| **+O$_2$ (1$^{st}$)** | $C_{10}H_{14}O_4$ (3.34718E+7) | $C_{10}H_{14}O_4$ (2.52885E+6) |
| **+O$_2$ (2$^{nd}$)** | $C_{10}H_{14}O_6$ (9.58108E+6) | $C_{10}H_{14}O_6$ (7.56393E+4) |
| **+O$_2$ (3$^{rd}$)** | $C_{10}H_{14}O_8$ (9.55306E+5) | $C_{10}H_{14}O_8$ (2.91182E+4) |
| **+O$_2$ (4$^{th}$)** | $C_{10}H_{14}O_{10}$ (1.00597E+4) | $C_{10}H_{14}O_{10}$ (not detected) |
| | $C_{10}H_{14}O_{12}$ (not detected) | -- |
| **Odd number of oxygen atom** | | |
| **3ydroperoxyl carbonyl** | $C_{10}H_{14}O_3$ (3.23297E+7) | $C_{10}H_{14}O_3$ (9.3999E+6) |
| **+O$_2$ (1$^{st}$)** | $C_{10}H_{14}O_5$ (2.27278E+7) | $C_{10}H_{14}O_5$ (1.17044E+5) |
| **+O$_2$ (2$^{nd}$)** | $C_{10}H_{14}O_7$ (4.04207E+6) | $C_{10}H_{14}O_7$ (7.17307E+4) |
| **+O$_2$ (3$^{rd}$)** | $C_{10}H_{14}O_9$ (1.92816E+5) | $C_{10}H_{14}O_9$ (not detected) |
| **+O$_2$ (4$^{th}$)** | $C_{10}H_{14}O_{11}$ (6.34129E+3) | -- |
| | $C_{10}H_{14}O_{13}$ (not detected) | -- |

The discussion has been reconsidered in the light of these clarifications (lines 315-332)

In addition, we have specified in the database, in the case of combustion, the chemical formulae consistent with an autoxidation mechanism and added a comment (lines 451-458):

JSR-FIA-HRMS data indicated many more sets of chemical formulas differing by two O-atoms in the range $C_{10}H_xO_{2-10}$ ($C_{10}H_{12}O_{2-10}$, $C_{10}H_{12}O_{3-9}$, $C_{10}H_{14}O_{2-10}$, $C_{10}H_{14}O_{3-9}$, $C_{10}H_{16}O_{2-8}$, $C_{10}H_{16}O_{3-9}$); see Supplementary database. Although at this stage one cannot prove these species were formed through autoxidation, their formulas are consistent with autoxidation products

**Comments:**

I disagree that if compounds, especially closed-shell species shown for instance in Table 3, are separated by two oxygens, they are directly connected by autoxidation. In other words, $C_{10}H_{16}O_6$ probably did not directly result in autoxidation of $C_{10}H_{16}O_4$ and so on.

**Answer:**

We agree with this comment. Indeed, only radicals are directly connected by addition O2 through autoxidation (Fig S3). However, $C_{10}H_{16}O_x$ species can derive from addition of OH on terpenes double bonds followed by O2 addition and autoxidation of products (Fig. S5).

**Comments:**

Nothing is mentioned about several oxidation steps (not autoxidation) or if those are expected in combustion (this was asked during review, but not expanded on by authors).

**Answer:**

The reactions presented in the introduction cover a set including autoxidation, but not limited to autoxidation:

R•+ O2 ⇄ ROO• (first O2-addition)

ROO• ⇄•QOOH (H-shift)

•QOOH +O2 ⇄ •OOQOOH (second O2-addition)

•OOQOOH ⇄ HOOQ•'OOH (H-shift)

HOOQ•'OOH +O2 ⇄ (HOO)2Q'OO• (third O2-addition)

(HOO)2Q'OO• ⇄ (HOO)2Q•"OOH (H-shift)

(HOO)2Q•"OOH +O2 ⇄ (HOO)3Q"OO• (fourth O-addition), etc.

The above reactions describe an autoxidation mechanism (cited in the Introduction). Those below are complementary and can occur under both 'atmospheric' and 'combustion' conditions (they are cited in the manuscript):

ROO• → R-HO + OH

ROO• + HOO• → ROOH + O2

ROO• + R'OO• → RO• + R'O• + O2 and $R_{-HO}$ + R'OH + O2

ROO• + R'OO• → ROOR' + O2)

QOOH → cyclic ether (QO) + OH.

Mechanisms are presented in the literature; we cite the paper of Wang et al. (2017) in PNAS where many reaction pathways are described, including autoxidation.

**Comments:**

The authors mentioned they added the database of detected molecules, but I did not find it in the revised Supplementary. It would have made a bit more clear, which compounds did not meet the criteria of "autoxidation" if we had a list of compounds with the marked selection.

**Answer:**

A database was added with all detected molecules for limonene and α-pinene (HESI, negative & positive modes). The molecules of interest cited in the article have been highlighted in red. The 78 molecules common to the four studies cited in the article and combustion experiments are present in the database.

**Comments:**

Since authors make a comprehensive review of the detected products, I do not find some of the replies related to compounds with formulas CxH4Oy and C21+HxOy sufficient. Also, I realized there are some CxH20Oy compounds. Since these formulas were identified in the authors' experiments, there should have been some explanation what those may be (are they artifacts or do they form in multiple oxidation steps, or there is another explanation?). Some of those species should not exist. Those are seen in Figure 2b, but also other figures and are actually discussed in terms of high DBE content, and something about their origin should have been mentioned.

**Answer:**

**Formulas C$_x$H$_4$O$_y$:** clarifications have been made in the text (lines 332-335).

Finally, some chemical formulas have no chemical relevance (CxH4Oy). These are probably artefacts linked to the characterization method (ionization mode, ions transfer, ions isolation in the c-trap, incorrect masse identification). We chose to leave these data, knowing that they would be discarded in the various subsequent comparisons.

**Formulas C$_{21}$+H$_x$O$_y$:** We have specified line 294-297 that the combinations of fragmentations (C<10) and accretions, observed in Figure 2a, form a continuous incrementation of the carbon number. We have added the explanation of the C=30 limit. This limit is probably due to the ionization or detection possibilities of the spectrometer.

The chemical formulas **C$_x$H$_{20}$O$_y$** cited and presented in figure 2b correspond to the C$_{10}$H$_{20}$O$_y$ formulas (in figure 2B, the carbon number is 10)

**Comments:**

Lines 254-256: I am still not sure how observed compounds with high DBE is a result of 'hydroperoxide and carboxyl groups along with the fact that a heated ionization source favors vaporization of low volatility compounds. This is the revised sentence, but authors did not provide justification.

**Answer:**

The increase in DBE results from the decrease of 2 hydrogens atoms or the increase of 1 carbon atoms. In this case, without carbonyl formation (while the number of oxygens increases), the increase in DBE would only result from the formation of vinyl or unsaturation groups. These conditions seem unlikely when the DBE is greater than 5, 10 or 15 under oxidation conditions. For this reason, we have used the term probable.

The volatility of a chemical compound is always favored by the rise in temperature. The modification of ESI sources to HESI has greatly contributed to the improvement of the ionisation of low volatile chemical compounds.

**Comments:**

Line 467. Where this D2O experiment shown?

**Answer:**

The results obtained by H/D exchanges with D2O and derivatization of carbonyls with 2,4-DNPH were added to the Fig S7 (supplements).

[Figure]

Fig. S7: D$_2$O exchanges for limonene and α-pinene (direct infusion, negative ionization mode) - top; and derivatization of carbonyls in limonene and α-pinene oxidation samples using 2,4-DNPH (UHPLC, positive mode) – bottom.

**Comments:**

Comparison of combustion to ozonolysis studies remains questionable as it should produce somewhat different autoxidation products. Some discussion if these compounds are expected to be the same in any circumstance would be useful.

**Answer**:

Indeed, "The subject is challenging because the oxidation chemistry of monoterpenes is highly complex and far from fully understood, either under atmospheric or cold flame conditions. It is clear that covering both regimes in all details in one paper is difficult to achieve".

We hope to have contributed some input to this question. It is clear that the database created will be a useful complement for those who want to continue this work.